# TUNING TIMESTEP-DISTILLED DIFFUSION MODEL USING PAIRWISE SAMPLE OPTIMIZATION

**Zichen Miao**[1], **Zhengyuan Yang**[2], **Kevin Lin**[2], **Ze Wang**[3]
**Zicheng Liu**[3], **Lijuan Wang**[2], **Qiang Qiu**[1]
[1]Purdue University, [2]Microsoft, [3]AMD

## ABSTRACT

Recent advancements in timestep-distilled diffusion models have enabled high-quality image generation that rivals non-distilled multi-step models, but with significantly fewer inference steps. While such models are attractive for applications due to the low inference cost and latency, fine-tuning them with a naive diffusion objective would result in degraded and blurry outputs. An intuitive alternative is to repeat the diffusion distillation process with a fine-tuned teacher model, which produces good results but is cumbersome and computationally intensive: the distillation training usually requires magnitude higher of training compute compared to fine-tuning for specific image styles. In this paper, we present an algorithm named pairwise sample optimization (PSO), which enables the direct fine-tuning of an arbitrary timestep-distilled diffusion model. PSO introduces additional reference images sampled from the current time-step distilled model, and increases the relative likelihood margin between the training images and reference images. This enables the model to retain its few-step generation ability, while allowing for fine-tuning of its output distribution. We also demonstrate that PSO is a generalized formulation which can be flexibly extended to both offline-sampled and online-sampled pairwise data, covering various popular objectives for diffusion model preference optimization. We evaluate PSO in both preference optimization and other fine-tuning tasks, including style transfer and concept customization. We show that PSO can directly adapt distilled models to human-preferred generation with both offline and online-generated pairwise preference image data. PSO also demonstrates effectiveness in style transfer and concept customization by directly tuning timestep-distilled diffusion models. The code is provided at: `https://github.com/ZichenMiao/Pairwise_Sample_Optimization`.

## 1 INTRODUCTION

Diffusion models (Ho et al., 2020; Song et al., 2020; Nichol & Dhariwal, 2021; Ho & Salimans, 2022; Karras et al., 2022; Rombach et al., 2022) have shown strong capabilities in generating high-fidelity images, marking a significant advancement in the field of generative modeling. Despite their impressive performance, a notable challenge is the high inference cost due to its iterative denoising nature. To address this issue, various methods are proposed to accelerate the sampling process of diffusion models, including improving the efficiency of samplers (Karras et al., 2022; Lu et al., 2022a;b; Liu et al., 2022; Zhao et al., 2024) and employing model distillation techniques (Song et al., 2023; Luo et al., 2023; Sauer et al., 2023; Xu et al., 2023; Lin et al., 2024; Sauer et al., 2024; Yin et al., 2024b;a; Ren et al., 2024; Kohler et al., 2024) to reduce the number of inference steps. Recent advancements in trajectory distillation methods and distribution matching techniques, often enhanced by adversarial learning at scale, have shown considerable promise in generating high-fidelity images in extremely low steps such as one to four steps.

Despite significant advancements in timestep-distilled diffusion models, it remains unclear how to effectively fine-tune or customize such distilled models. Naively tuning the distilled model with diffusion loss will make the generation results blurry, as shown in Figure 1 (b). An alternative approach is to fine-tune or customize the original diffusion model, and then repeat the diffusion distillation process to create a distilled model variant. However, the large computation cost of diffusion distillation, when compared with the customization training used for applications (*cf.*, *3840* A100 GPU

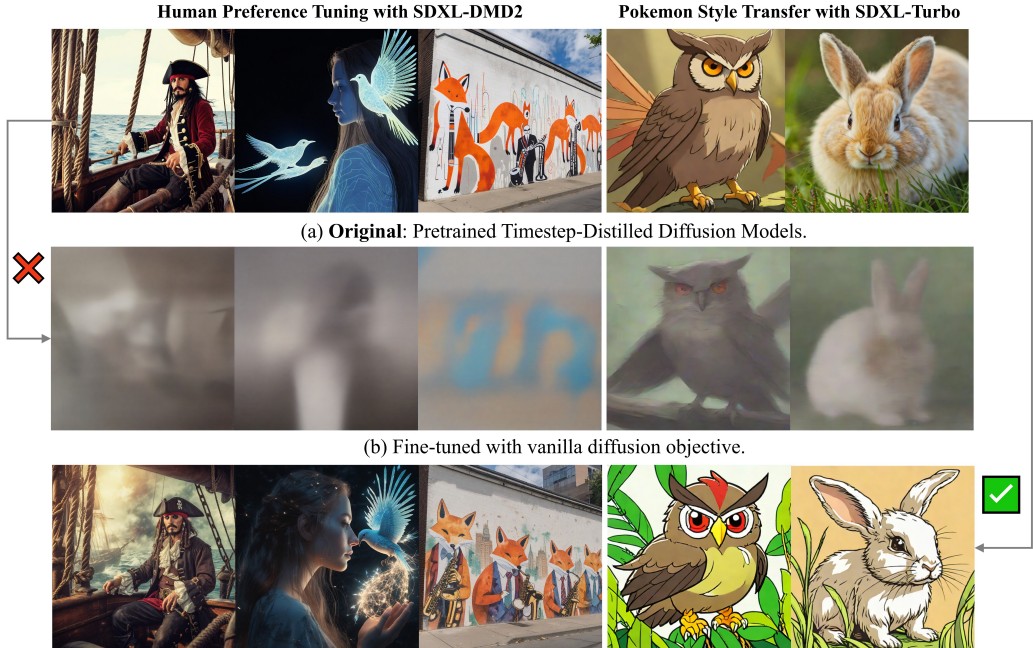

**Human Preference Tuning with SDXL-DMD2**    **Pokemon Style Transfer with SDXL-Turbo**

(a) **Original**: Pretrained Timestep-Distilled Diffusion Models.

(b) Fine-tuned with vanilla diffusion objective.

(c) **PSO** : Fine-tuned with our pairwise sample optimization.

Figure 1: Illustration of images sampled from: (a) original timestep-distilled diffusion models, (b) Fine-tuned distilled model with diffusion objective (Ho et al., 2020), and (c) Fine-tuned distilled models with our PSO, and It can be seen that simply tuning distilled models with the vanilla diffusion loss leads to blurry, degraded generation, while our method can steer the distilled model toward better alignment with human preference & prompts, and style-transferred generation. Prompt from left to right: *A Pirate in a Pirateship.// a woman with long hair next to a luminescent bird.// Photograph of a wall along a city street with a watercolor mural of foxes in a jazz band.// A stern-faced, brown-feathered owl Pokémon with a leaf-shaped crown and piercing red eyes.// A cute rabbit.*

hours for SDXL-DMD2 (Yin et al., 2024a) and *0.25* A100 GPU hours for Dreambooth concept customization (Ruiz et al., 2023)), often makes such distilled model tuning approach less feasible.

In this paper, we present an effective algorithm named pairwise sample optimization (PSO), designed to directly tuning a timestep-distilled diffusion model for different user preferences or customizing it towards new image domains. PSO maximizes the likelihood ratio between a given pair of target and reference images, where the target image is sampled from the distribution we wish to tune the distilled model towards, and the reference image is sampled from the generative distribution of the tuned distilled models on-the-fly. By optimizing this relative likelihood objective, PSO effectively aligns the timestep-distilled models with target distributions while best preserving its original few-step generation ability learned through diffusion distillation. Our formulation of PSO can be flexibly adapted to both offline and online settings, and unifies various prior works (Wallace et al., 2024; Yang et al., 2024a) as special cases to our framework.

We validate the effectiveness of the proposed method across a broad range of tasks, including human preference tuning, style transfer, and concept customization. For preference tuning and style transfer, we construct or directly utilize target-reference images with the given prompts, *e.g.*, win-lose pairs from the user preference data (Kirstain et al., 2023), and adopt our offline PSO objective to tune the distilled models. We also examine our online objective in preference tuning, where we sample pairs of images from distilled model and use a reward mode to assign the label of target or reference. For concept customization, we use the given concept images as target and sample reference images from the tuned model. In all tasks and settings, PSO enables the lightweight fine-tuning of arbitrary distilled models while preserving their few-step generation capability, as shown in Figure 1 (c).

We evaluate our method on various datasets and benchmarks. For human preference tuning, we benchmark PSO on the standard Pick-a-Pic (Kirstain et al., 2023) and PartiPrompts (Yu et al., 2022)

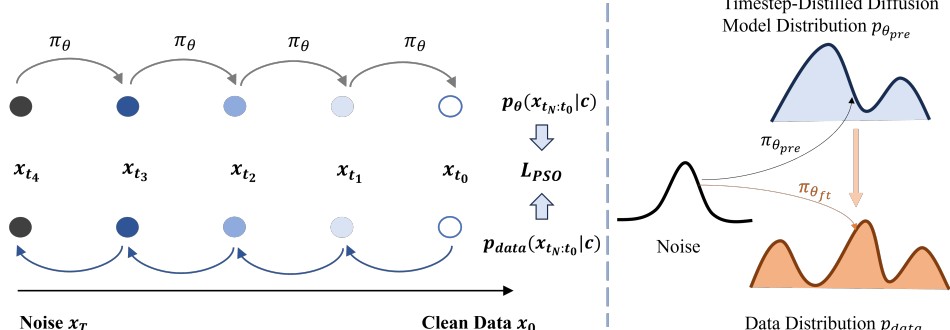

Figure 2: Demonstration of the proposed pairwise sample optimization. To tune the generative distribution $p_\theta$ to $p_{data}$, we sample a pair of images together with their trajectories of the same prompt, where we adopt our Markov Decision Process (MDP) formulation for the timestep-distilled diffusion model to efficiently sample the backward denoising trajectories $\{x_{t_n}^\rho\}$, while sampling $\{x_{t_n}^\tau\}$ from data via the forward diffusion process. The sampled trajectories are then sent to the final objective to move the generation trajectory aligned with the forward process from $p_{data}$.

datasets. We also evaluate distilled model fine-tuning by experimenting the image style transfer task with the Pokemon dataset (Pinkney, 2022) and the concept customization task with the exeamples in Dreambooth (Ruiz et al., 2023). We observe that PSO shows comparable performance to the oracle results of fine-tuning a new teacher model and conduct the diffusion distillation training, while being much less computationally expensive. Our approach also significantly outperforms other designed baselines achieved via the use of LoRA parameters.

Our contributions are summarized as follows.

- We present pairwise sample optimization (PSO) that can directly tune timestep-distilled diffusion models toward a new output distribution for various customization applications.
- We show that PSO is a generalized formulation that covers both offline and online preference optimization scenarios, encompassing previous works, such as DiffusionDPO (Wallace et al., 2024) and D3PO (Yang et al., 2024a) as special cases, which can be flexibly adapted to various settings and tasks.
- Extensive of experiments demonstrate the effectiveness of PSO in human preference tuning, as well as other domain transfer tasks such as style transfer and concept customization.

## 2 Method

In this section, we present our method of pair-wise sample optimization (PSO) for directly fine-tuning timestep-distilled diffusion models. We first show the PSO formulation with both reference and data trajectories, and then extend it to both online and offline settings.

### 2.1 Pairwise Sample Optimization for Timestep-Distilled Diffusion Models

Consider a timestep-distilled diffusion model $p_\theta(x_0|c)$ that samples clean images $x_0$ conditioned on the prompt $c$ from initial noise $x_T \sim N(0, I)$ with very few timesteps $N, N = 1 \sim 4$, we denote its sampling trajectory as $\{x_{t_N}, x_{t_{N-1}}, ..., x_{t_1}, x_{t_0}\}$, where $t_N = T, t_0 = 0$. Our target is to fine-tune the model towards the given target distribution $p_{data}(x_0^\tau, c)$, i.e., maximizing the log-likelihood $\log p_\theta(x_0^\tau|c)$, where $x_0^\tau \sim p_{data}$. Directly adopting the DDPM (Ho et al., 2020) formulation and minimizing the diffusion objective $||\epsilon_\theta(x_{t_n}^\tau, t) - \epsilon||^2$ leads to blurry, degraded generation. To mitigate this issue, we introduce the reference sample $x_0^\rho$ from the current model $p_\theta$ and recast the optimization as maximizing the relative likelihood between the target and reference samples. Intuitively, by maximizing $\log \frac{p_\theta(x_0^\tau|c)}{p_\theta(x_0^\rho|c)}$, we can steer the generative distribution $p_\theta$ towards the data distribution $p_{data}$. Meanwhile, we avoid directly minimizing the diffusion objective, which can potentially help preserve its ability on few-step generation.

Formally, given a pair of positive and negative images with the same prompt $c$, $x_0^\tau \sim p_{data}(x_0^\tau|c)$, $x_0^\rho \sim p_\theta(x_0^\rho|c)$, we take inspiration from direct preference optimization (Rafailov et al., 2024) and maximize the margin between $\log p_\theta(x_0^\tau|c)$ and $\log p_\theta(x_0^\rho|c)$, regularized by the pre-trained timestep-distilled model $\theta_{pre}$,

$$\mathcal{L} = -\mathbb{E}_{(x_0^\tau, x_0^\rho, c)}\left[\log \sigma\left(\beta \log \frac{p_\theta(x_0^\tau|c)}{p_{\text{pre}}(x_0^\tau|c)} - \beta \log \frac{p_\theta(x_0^\rho|c)}{p_{\text{pre}}(x_0^\rho|c)}\right)\right], \tag{1}$$

where $\sigma$ denotes the sigmoid function, $\beta$ is the regularization weight, $p_{\text{pre}} = p_{\theta_{\text{pre}}}$ denotes the pre-trained model.

In diffusion models, obtaining $p_\theta(x_0|c)$ requires integrating over intermediate states which is costly. Instead, we consider the whole sampling trajectory $\{x_{t_n}|c\}_{n=0}^N$, and maximize the margin between their trajectory joint likelihoods. Specifically, as shown in Figure 2, we sample the data trajectory $\{x_{t_0:t_N}^\tau|c\}$ with diffusion forward process, and sample the reference trajectory $\{x_{t_0:t_N}^\rho|c\}$ with the generative reverse process,

$$p_{data}(x_{t_0:t_N}^\tau|c) = p_{data}(x_{t_0}^\tau|c)\prod_n q(x_{t_n}^\tau|x_{t_{n-1}}^\tau)$$
$$p_\theta(x_{t_0:t_N}^\rho|c) = p_\theta(x_{t_N}^\rho|c)\prod_n p_\theta(x_{t_{n-1}}^\rho|x_{t_n}^\rho, c). \tag{2}$$

Substitute the trajectory formulation into Eq. 1, and we obtain our PSO objective below,

$$\mathcal{L}_{\text{PSO}} = -\mathbb{E}_{(x_{0:t_N}^\tau, x_{0:t_N}^\rho, c)}\left[\log \sigma\left(\beta \sum_n \left(\log \frac{p_\theta(x_{t_{n-1}}^\tau|x_{t_n}^\tau, c)}{p_{\text{pre}}(x_{t_{n-1}}^\tau|x_{t_n}^\tau, c)} - \log \frac{p_\theta(x_{t_{n-1}}^\rho|x_{t_n}^\rho, c)}{p_{\text{pre}}(x_{t_{n-1}}^\rho|x_{t_n}^\rho, c)}\right)\right)\right]$$
$$= -\mathbb{E}_{(x_{0:t_N}^\tau, x_{0:t_N}^\rho, c)}\left[\log \sigma\left(\beta \sum_n \left(D_{\text{KL}}[q(x_{t_{n-1}}^\tau|x_{t_n,t_0}^\tau) \| p_{\text{pre}}(x_{t_{n-1}}^\tau|x_{t_n}^\tau, c)]\right.\right.\right.$$
$$\left.\left.\left.- D_{\text{KL}}[q(x_{t_{n-1}}^\tau|x_{t_n,t_0}^\tau) \| p_\theta(x_{t_{n-1}}^\tau|x_{t_n}^\tau, c)] - \log \frac{p_\theta(x_{t_{n-1}}^\rho|x_{t_n}^\rho, c)}{p_{\text{pre}}(x_{t_{n-1}}^\rho|x_{t_n}^\rho, c)}\right)\right)\right]. \tag{3}$$

The derivation is provided in Appendix A.

To expand the joint likelihood terms, we need to define the transition kernel $p_\theta(x_{t_{n-1}}|x_{t_n}, c)$. Inspired by previous works (Black et al., 2023; Yang et al., 2024a), we formulate the denoising sampling process of timestep-distilled models as a Markov Decision Process (MDP).

**MDP Formulation for Timestep-Distilled Diffusion Models.** Let $\epsilon_\theta(x_{t_n}, t_n, c)$ denote the timestep-distilled diffusion model, which predicts $x_0$ as $f_\theta(x_{t_n}, t_n, c) = f(\epsilon_\theta(x_{t_n}, t_n, c), x_{t_n}, t_n)$ given the corresponding noisy latent $x_{t_n}$ (Song et al., 2023; Yin et al., 2024b;a). In the iterative few-step sampling process, the distilled model first predicts $x_0$ at $t_n$, and then add noise back to noise-level $t_{n-1}$,

$$x_{t_{n-1}} = \sqrt{\bar{\alpha}_{t_{n-1}}} f_\theta(x_{t_n}, t_n, c) + \sqrt{1 - \bar{\alpha}_{t_{n-1}}} z, z \sim N(0, I), \tag{4}$$

where $\bar{\alpha}$'s are the forward process coefficients. The Markov Decision Process for distilled models can be then formulated as,

$$s_n = (x_{t_n}, t_n), \ a_n = x_{t_{n-1}}, \ P(s_{n+1}|s_n, a_n) = \delta(x_{t_{n-1}}, t_{n-1}, c)$$
$$\pi_\theta(a_n|s_n) = N(\sqrt{\bar{\alpha}_{t_{n-1}}} f_\theta(x_{t_n}, t_n, c), 1 - \bar{\alpha}_{t_{n-1}}) = N(\mu_\theta(x_{t_n}, t_n, c), \sigma_{t_n}^2 I),$$

where $s_n, a_n$ denote the state and action, $P(s_{n+1}|s_n, a_n)$ denotes the transition kernel, $\delta$ is the Dirac function, and $\pi_\theta$ is the policy. For the last timestep, the distilled model directly predict the clean image without noise added. This makes the final timestep a deterministic transition, $p_\theta(a_N|s_N) = \delta(f(x_{t_1}, t_1, c))$, so we remove the last time step from the training. With this formulation, we have $p_\theta(x_{t_{n-1}}|x_{t_n}) = \pi_\theta(a_n|x_n)$.

**Final PSO Loss.** By plugging the MDP action-state conditional distribution, we obtain the objective for our pairwise sample optimization,

$$\mathcal{L}_{\text{PSO}} = -\mathbb{E}\Bigg[ \log \sigma \Bigg( - \beta \cdot \sum_{n=2}^{N} \Big( \big( ||\epsilon^\tau - \epsilon_\theta(x_{t_n}^\tau, t_n, c)||^2 - ||\epsilon^\tau - \epsilon_{\text{pre}}(x_{t_n}^\tau, t_n, c)||^2 \big)$$
$$- \frac{1}{2\sigma_{t_n}^2} \Big( ||x_{t_{n-1}}^\rho - \mu_\theta(x_n^\rho, t_n, c)||^2 - ||x_{t_{n-1}}^\rho - \mu_{\text{pre}}(x_n^\rho, t_n, c)||^2 \Big) \Big) \Bigg) \Bigg]. \tag{5}$$

The derivation is provided in Appendix A. We provide Alg. 1 for better illustration.

## 2.2 OFFLINE AND ONLINE EXTENSION

Our pairwise sample optimization can be extended to online and offline settings as follows,

**Offline PSO.** In practice, we may have pre-sampled $\{x_{t_0}^\rho\}$ from models that have similar generation distribution with $p_\theta(\cdot|c)$, which removes the need for reverse sampling $\{x_{t_n}^\rho\}$ from $p_\theta$ again. In this case, we extend our PSO formulation by approximating $p_\theta(x_{t_1:t_N}^\rho|x_0^\rho, c)$ with the forward trajectory $q(x_{t_1:t_N}^\rho|x_0^\rho)$. Specifically, we approximate the per-step transition kernel $p_\theta(x_{t_{n-1}}^\rho|x_{t_n}^\rho, c)$ with $\int p(x_{t_0}^\rho|c)q(x_{t_{n-1}}^\rho|x_{t_n}, x_{t_0})$, and we obtain our offline PSO objective,

$$\mathcal{L}_{\text{PSO-Offline}} = -\mathbb{E}\Bigg[ \log \sigma \Bigg( - \beta \cdot \sum_{n=2}^{N} \Big( ||\epsilon^\tau - \epsilon_\theta(x_{t_n}^\tau, t_n, c)||^2 - ||\epsilon^\tau - \epsilon_{\text{pre}}(x_{t_n}^\tau, t_n, c)||^2$$
$$- ||\epsilon^\rho - \epsilon_\theta(x_{t_n}^\rho, t_n, c)||^2 + ||\epsilon^\rho - \epsilon_{\text{pre}}(x_{t_n}^\rho, t_n, c)||^2 \Big) \Bigg) \Bigg], \tag{6}$$

which can be seen as a variant of the offline Diffusion-DPO (Wallace et al., 2024). We provide Alg. 3 for better illustration.

**Online PSO.** There are also cases where we decide the data and the reference distribution in an online manner, e.g., with a reward model. In this case, we can substitute the forward trajectory in Eq. 3 with the generative trajectory,

$$\mathcal{L}_{\text{PSO-Online}} = -\mathbb{E}\Bigg[ \log \sigma \Bigg( - \frac{\beta}{2\sigma_{t_n}^2} \cdot \sum_{n=2}^{N} \Big( ||x_{t_{n-1}}^\tau - \mu_\theta(x_n^\tau, t_n, c)||^2 - ||x_{t_{n-1}}^\tau - \mu_{\text{pre}}(x_n^\tau, t_n, c)||^2$$
$$- ||x_{t_{n-1}}^\rho - \mu_\theta(x_n^\rho, t_n, c)||^2 + ||x_{t_{n-1}}^\rho - \mu_{\text{pre}}(x_n^\rho, t_n, c)||^2 \Big) \Bigg) \Bigg], \tag{7}$$

which can be seen as a variant of the online DPO for diffusion models (Yang et al., 2024a). We provide Alg. 2 for better illustration.

## 3 EXPERIMENTS

In this section, we validate our method with multiple tasks and experiments. First, we demonstrate the effectiveness of our proposed PSO in the human-preference tuning task. We then present results on other general fine-tuning tasks for timestep-distilled diffusion models with PSO, including style transfer and concept customization. Our experiments cover various main-stream timestep-distillation methods, including Distribution Matching (DMD) (Yin et al., 2024a;b), Adversarial Distillation (ADD) (Sauer et al., 2023), and Latent Consistency Models (LCM) (Luo et al., 2023). We

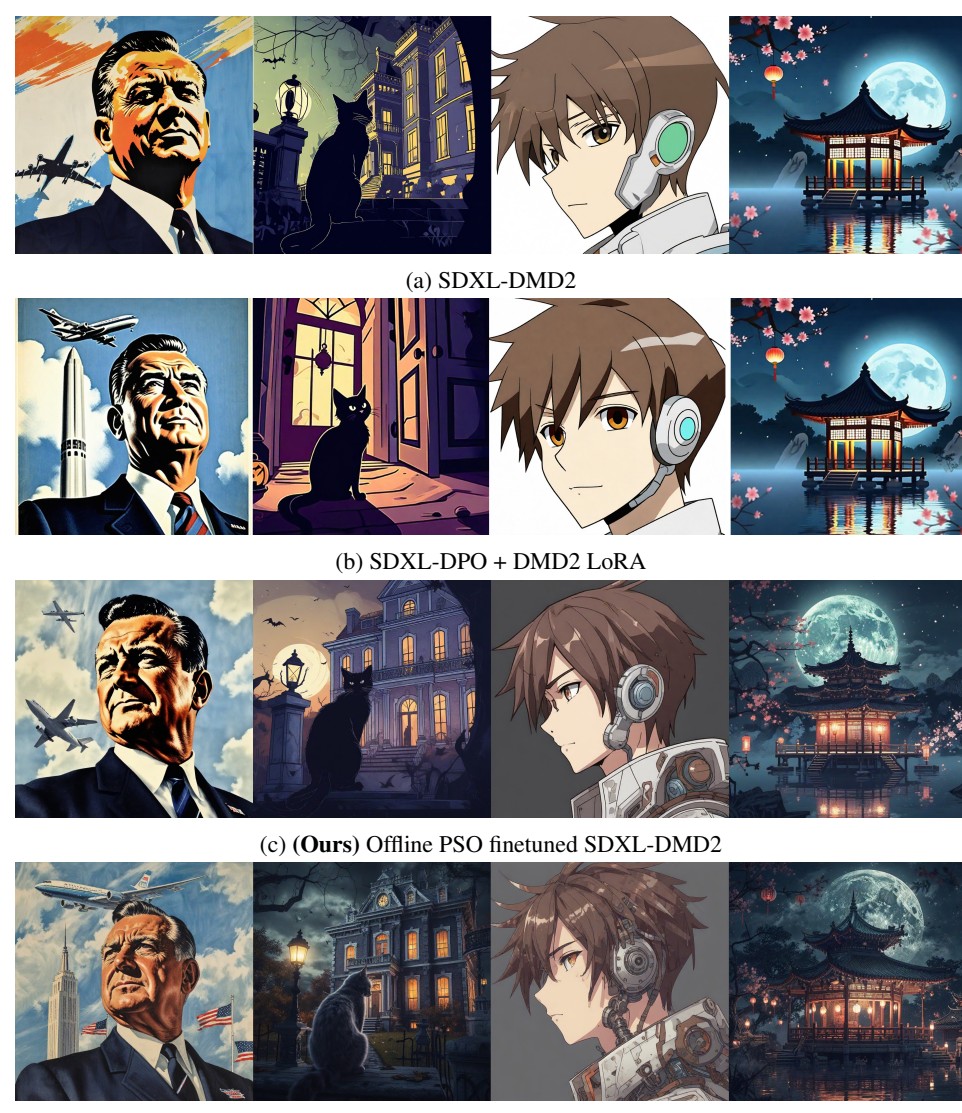

(a) SDXL-DMD2

(b) SDXL-DPO + DMD2 LoRA

(c) **(Ours)** Offline PSO finetuned SDXL-DMD2

(d) **(Ours)** Online PSO finetuned SDXL-DMD2

Figure 3: Human preference tuning Results with PSO on 4-step SDXL-DMD2. Compared with baseline SDXL-DMD2 (sub-figure (a)), SDXL-DPO with DMD2 LoRA (sub-figure (b)) exhibits a slightly degraded generation quality. Rather, SDXL-DMD2 with both our offline and online PSO objectives (sub-figures (c) and (d) respectively) demonstrates substantial improvement in visual appeal, prompt following, and details generation. Prompts from left to right: *The official portrait of an authoritarian president of an alternate America in 1960, in the style of pan am advertisements, looking up, jet age.// A curious cat exploring a haunted mansion.// A profile picture of an anime boy,* **half robot***, brown hair.//On the Mid-Autumn Festival, the bright full moon hangs in the night sky. A quaint pavilion is illuminated by dim lights, resembling a beautiful scenery in a painting. Camera type: close-up. Camera lenstype: telephoto. Time of day: night. Film type: ancient style. HD.*

use the distilled versions of SDXL (Podell et al., 2023) with these methods as our base models, i.e., SDXL-DMD2 (4 step) (Yin et al., 2024a), SDXL-Turbo (Sauer et al., 2023), and SDXL-LCM (Luo et al., 2023).

## 3.1 PSO FOR HUMAN-PREFERENCE FINE-TUNING

**Human Preference Tuning with Offline PSO.** We first examine the effectiveness of PSO in human-preference fine-tuning in the offline setting. Specifically, we consider pre-sampled reference image samples $\{x_0^\rho|c\}$ as discussed in Sec. 2.2, which are negative samples in the human preference dataset in this case, and the data samples $\{x_0^\tau|c\}$ are the preferred images in the dataset. We train all

Table 1: Human preference tuning Results of SDXL-DMD2.

| Dataset | Method | Inference Steps | PickScore | CLIP Score | ImageReward | Aesthetic Score |
|---|---|---|---|---|---|---|
| Pickapic Test | SDXL | 50 | 22.30 | 0.3713 | 0.8556 | 6.060 |
| | SDXL-DPO(Wallace et al., 2024) | 50 | 22.60 | **0.3787** | **1.0075** | 6.040 |
| | SDXL-SFT | 50 | 22.25 | 0.3693 | 0.8665 | 5.921 |
| | SDXL-RPO (Gu et al., 2024) | 50 | 22.65 | 0.3723 | 0.9623 | 6.012 |
| | SDXL-SPO (Liang et al., 2024) | 50 | 22.70 | 0.3527 | 0.9417 | **6.283** |
| | SDXL-MaPO (Hong et al., 2024) | 50 | 22.50 | 0.3735 | 0.9481 | 6.170 |
| | SDXL-DMD2 | **4** | 22.35 | 0.3679 | 0.9363 | 5.937 |
| | SDXL-DPO + DMD2-LoRA | **4** | 22.20 | 0.3673 | 0.9287 | 5.759 |
| | Offline-PSO w/ SDXL-DMD2 | **4** | 22.46 | **0.3690** | 0.9381 | 5.994 |
| | Online-PSO w/ SDXL-DMD2 | **4** | **22.73** | 0.3671 | **0.9773** | 6.077 |
| Parti-Prompts | SDXL | 50 | 22.77 | 0.3607 | 0.9142 | 5.750 |
| | SDXL-DPO | 50 | 22.92 | **0.3674** | 1.1180 | 5.795 |
| | SDXL-SFT | 50 | 22.85 | 0.3610 | 0.8565 | 5.675 |
| | SDXL-RPO (Gu et al., 2024) | 50 | 22.98 | 0.3670 | 1.0770 | 5.872 |
| | SDXL-SPO (Liang et al., 2024) | 50 | 23.27 | 0.3428 | 1.0668 | **6.083** |
| | SDXL-MaPO (Hong et al., 2024) | 50 | 22.81 | 0.3661 | 1.0315 | 5.912 |
| | SDXL-DMD2 | **4** | 22.99 | 0.3607 | 1.0713 | 5.671 |
| | SDXL-DPO + DMD2-LoRA | **4** | 22.76 | 0.3644 | 1.0638 | 5.513 |
| | Offline-PSO w/ SDXL-DMD2 | **4** | 23.07 | **0.3649** | 1.0964 | 5.715 |
| | Online-PSO w/ SDXL-DMD2 | **4** | **23.29** | 0.3634 | **1.1702** | **5.836** |

the base timestep-distilled diffusion models with the Pick-a-Picv2 (Kirstain et al., 2023) dataset. After removing the pairs with tied preference labels, we end up with 851,293 pairs, with 58,960 unique prompts. We adopt the PSO-Offline objective in Eq. 6 and fine-tune all our base timestep-distilled models. The training details and hyperparameters are provided in Appendix B.1.

**Human Preference Tuning with Online PSO.** We also validate the proposed method with the online human-preference tuning task, where we decide sampled data belonged to $p_{data}$ or $p_\theta$ throughout the training based on the human preference model PickScore (Kirstain et al., 2023). We tune all the base distilled models with the PSO objective in Eq. 7 with a subset of training prompts from Pick-a-Pic v2 (Kirstain et al., 2023). All the other training details are provided in Appendix B.1.

**Evaluation and Benchmarks.** We use the Pick-a-Pic test set and PartiPrompts (Yu et al., 2022) as the evaluation benchmarks, where we fix the random seed for all models and generate images for all prompts in the dataset. As for the evaluation metrics, we adopt PickScore (Kirstain et al., 2023), CLIP score (Radford et al., 2021), ImageReward (Xu et al., 2024), and Aesthetic score following Rafailov et al. (2024). We benchmark distilled models fine-tuned with PSO with various benchmarks and settings. For DMD (Yin et al., 2024a) distillation method, We compare our fine-tuned SDXL-DMD2 with un-distilled multi-step(25 steps) SDXL models, SDXL with offline human preference tuning, SDXL-DPO (Rafailov et al., 2024). We also benchmark with timestep-distilled models, SDXL-DMD2, SDXL with DMD2 LoRA applied, and SDXL-DPO with DMD2 LoRA applied. We adopt the similar evaluation settings for SDXL-LCM, as detailed in Appendix B.1. For SDXL-Turbo, we fine-tune the model with 4 steps ($N = 4$), and we evaluate the results on both 1-step and 4-step settings to show the timestep generalization ability of the proposed method.

**Results.** Table 1 shows the experiment results with DMD2 (Yin et al., 2024a) as the base model. Directly applying the DMD2 LoRA to human-preference fine-tuned SDXL-DPO (row with SDXL-

Table 2: Human preference tuning results of SDXL-Turbo.

| Dataset | Method | Inference Steps | PickScore | CLIP Score | ImageReward | Aesthetic Score |
|---|---|---|---|---|---|---|
| Pickapic Test | SDXL-Turbo-4step | 4 | 22.22 | 0.3610 | 0.9300 | 5.987 |
| | Offline-PSO w/ SDXL-Turbo-4step | 4 | 22.40 | 0.3634 | 0.9695 | 6.029 |
| | Online-PSO w/ SDXL-Turbo-4step | 4 | **22.71** | **0.3647** | 0.9882 | **6.157** |
| | SDXL-Turbo-1step | 1 | 22.29 | 0.3642 | 0.8830 | 6.061 |
| | Offline-PSO w/ SDXL-Turbo-1step | 1 | 22.40 | **0.3663** | 0.9073 | 6.072 |
| | Online-PSO w/ SDXL-Turbo-1step | 1 | **22.62** | 0.3661 | 0.9113 | **6.137** |
| Parti-Prompts | SDXL-Turbo-4-step | 4 | 22.88 | 0.3594 | 1.0173 | 5.709 |
| | Offline-PSO w/ SDXL-Turbo-4step | 4 | 22.96 | **0.3642** | 1.0509 | 5.746 |
| | Online-PSO w/ SDXL-Turbo-4step | 4 | **23.23** | 0.3632 | **1.0893** | **5.837** |
| | SDXL-Turbo-1-step | 1 | 22.78 | 0.3596 | 0.9246 | 5.706 |
| | Offline-PSO w/ SDXL-Turbo-1step | 1 | 22.86 | 0.3631 | 0.9664 | 5.755 |
| | Online-PSO w/ SDXL-Turbo-1step | 1 | **22.96** | **0.3632** | **0.9762** | **5.795** |

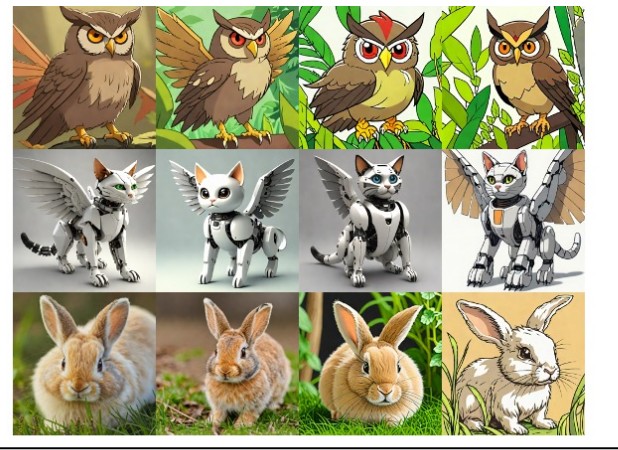 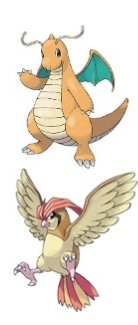

Original Generation          Fine-tuning steps          Target Pokemon Style

Figure 4: Experiments on tuning SDXL-Turbo for style transfer with PSO. The proposed method effectively tunes the distilled model to generate images following the targeted style. Prompts from top to bottom row: *A stern-faced, brown-feathered owl Pokémon with a leaf-shaped crown and piercing red eyes stands ready for a battle.// robotic cat with wings.// A cute bunny rabbit.*

DPO + DMD2 LoRA) achieves slightly inferior results compared with the SDXL-DMD2 baseline. It suffers from a drop in PickScore (drops from 22.35 to 22.20 in PickaPic-test, and 22.99 to 22.77 in Parti-Prompts) and Aesthetic Score (drops from 5.937 to 5.759 in PickaPic-test, and 5.671 to 5.513 in Parti-Prompts). The performance drop indicates the need to redo the diffusion distillation from SDXL-DPO to obtain a distilled model better aligned with human preference, which is costly ($\sim$ 4000 A100 GPU hours (Yin et al., 2024a)) for each fine-tuning. In contrast, our method achieves superior results compared with the SDXL-DMD2 baseline. For both Pick-a-Pic test set and Parti-Prompts, our Offline-PSO fine-tuned model achieves higher results in all metrics compared with baselines, showing its dominance in visual quality and human preference. For online-PSO where both the data and reference sets are updated together with the model, it achieves the best results in PickScore, ImageReward, and Aesthetic Score among all distilled models. Furthermore, its performance can even surpass the performance of multi-step fine-tuned SDXL-DPO, which further highlights the effectiveness of our method. For instance, our method achieves PickScore of 22.73 and 23.29 in two datasets, while SDXL-DPO has scores of 22.60 and 22.92, Qualitatively, as shown in Figure 3, both PSO-Offline and PSO-Online fine-tuned models demonstrate better improvement in image quality in terms of prompt following, visual appeal, and the intricacy of details within each image. Moreover, Online PSO fine-tuned SDXL-DMD2 shows notable improvement over other baselines in terms of aesthetic appeal and high-frequency details, underscores the effectiveness of the proposed method.

Table 2 reports the results on tuning SDXL-Turbo (Sauer et al., 2023). We observe a similar trend as in Table 1, where both Offline-PSO and Online-PSO achieves superior results of all metrics in 4-step generation compared with the base distilled SDXL-Turbo. Moreover, our fine-tuned model also achieves superior results in 1-step generation, showcasing the generalization ability of the proposed method. We provide qualitative results of SDXL-Turbo, along with the results on SDXL-LCM in Appendix B.1, where the proposed consistently surpass baselines in terms of visual appeal and intricate detail generation.

## 3.2 PSO Fine-tuning for Style Transfer

In this section, we demonstrate that our PSO can steer the timestep-distilled models towards a specific generation style. We select the Pokemon dataset (Pinkney, 2022) that contains Pokemon-style image-text pairs, and we utilize the proposed PSO to have the model generate images following the Pokemon style in the dataset. Specifically, we are interested in distilled models that has no equivalent distillation LoRA, so we select SDXL-Turbo (Sauer et al., 2023). We adopt the PSO objective in Eq. 3, and sample the reverse generative trajectories in each step. Other tarining details and hyperparameters are provided in Appendix B.2. We show the generation results of the Offline-PSO

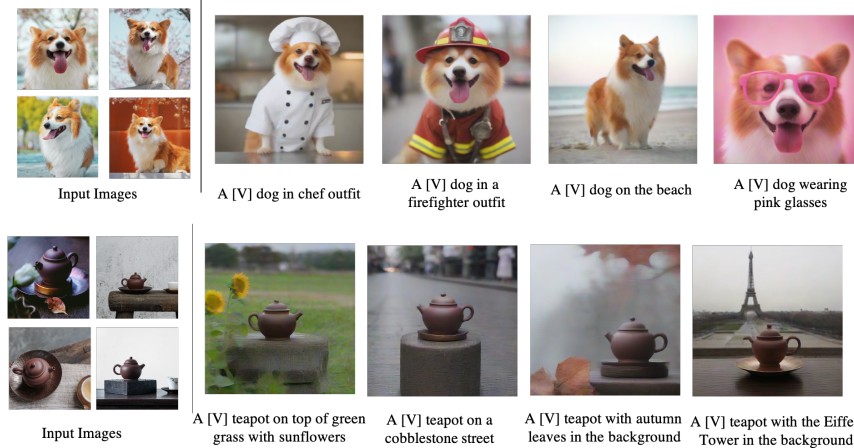

Figure 5: Experiments of tuning SDXL-Turbo for concept customization with PSO. The proposed method can effectively tune SDXL-Turbo to generate images that contain the given objects.

fine-tuned SDXL-Turbo in Figure 4. We observe that along the fine-tuning progress, the generated images gradually shift from the original generative distribution to match the style of the given Pokemon style.

## 3.3 PSO Fine-tuning for Concept Customization

In this section, we validate the PSO fine-tuning method with concept customization with SDXL-Turbo (Sauer et al., 2023). Following Dreambooth (Ruiz et al., 2023), we aim to tune the model to associate the special token with the given object or concept represented in few (around 5) images, and make it generalize on unseen environments. To minimize the fine-tuning costs, we adopt the offline-PSO setting. We also observe that in omitting the reference model and add prior preservation loss leads to performance gain in concept customization. Training details and hyperparameters are given in Appendix B. We show the qualitative results in Figure 5, where we can see the PSO fine-tuned SDXL-Turbo generate images that contain the exact the same object shown in the training input images. This concept customization results further demonstrate the effectiveness of PSO and its generalization among different tasks. More results are provided in Appendix B.3. We also provide quantitative results in Table 4, where our method shows superior results than vanilla tuning of SDXL-Turbo following Ruiz et al. (2023), and comparable results with the multi-step customization baseline (Dreambooth w/ SDXL).

## 4 Related Works

### 4.1 Diffusion Model Distillation

Knowledge distillation, initially proposed for neural network compression (Hinton, 2015; Cho & Hariharan, 2019; Patel et al., 2023), has become a powerful technique for transferring knowledge from a complex teacher model to a simpler student model while maintaining comparable performance. To accelerate the inference speed of the diffusion models, various timestep distillation methods have been proposed to transfer the generative distribution from a multi-step teacher model to a few-step student model. This is achieved by either distilling the PF-ODE from the pre-trained multi-step diffusion model (Song et al., 2023; Salimans & Ho, 2022; Kim et al., 2023), or matching the denoising distribution with an explicit analytical distribution matching loss (Yin et al., 2024b;a) or by learning an implicit discriminator (Sauer et al., 2023; Xu et al., 2023; Sauer et al., 2024; Lin et al., 2024). There are also recent works that combine those approaches to further boost the generation quality (Ren et al., 2024; Chadebec et al., 2024). However, the timestep-distilled models generally lose the ability to predict the local score, indicating that they can not be directly fine-tuned with the original score-matching loss towards a target distribution, as shown in Figure 1. In this work,

we tackle this problem with the incorporation of the reference samples, and formulate the pairwise sample optimization objective to directly fine-tune timestep-distilled diffusion models.

## 4.2 PREFERENCE OPTIMIZATION FOR DIFFUSION MODELS

To control the generation behavior of the diffusion models, researchers propose the markov decision process (MDP) formulation and utilize reinforcement learning (RL) to tune the multi-step diffusion modelsBlack et al. (2023); Fan et al. (2023) to generate images that have higher aesthetic values, better prompt-following ability, alignment with human preference, and diversity (Kirstain et al., 2023; Xu et al., 2024; Liu et al., 2024a; Miao et al., 2024). Inspired by Rafailov et al. (2024), researchers have further derived direct preference optimization (DPO) from the RL formulation for diffusion models with both online (Yang et al., 2024a;b; Liang et al., 2024) and offline settings (Wallace et al., 2024). Yuan et al. (2024) takes a step further by using SFT data as the preferred images while sampling negative images from the model progressively, However, all those methods are designed for tuning base multi-step diffusion models, which are not applicable to distilled few-step models, and they only focus on human preference tuning. In this work, we formulate the MDP for timestep-distilled models for direct fine-tuning, and show the effectiveness of the proposed method in fine-tuning tasks including style transfer and concept customization.

## 4.3 DIFFUSION MODELS TUNING FOR STYLE TRANSFER AND CUSTOMIZATION

Given the base multi-step diffusion models, a large corpus of works has focused on fine-tuning the models toward a specific style (Huang et al., 2022; Wang et al., 2023; Sohn et al., 2023; Zhang et al., 2023), and customizing the models with a given concept (Ruiz et al., 2023; Xiong et al., 2024; Song et al., 2024b; Kumari et al., 2023). However, those methods are specifically designed for the corresponding task, and are general limited for multi-step base diffusion models. In our work, we target at directly tuning timestep-distilled diffusion models with a unified objective that can be also applied to human-preference tuning.

## 5 CONCLUSION

In this work, we presented pairwise sample optimization (PSO) for directly fine-tuning timestep-distilled diffusion models. By constructing pairs of data and reference images and utilizing preference optimization to increase the likelihood margin between them, PSO enables manipulation of the model's generative distribution while preserving its few-step generation capability. We reformulated preference optimization for timestep-distilled models and demonstrated its effectiveness in aligning model outputs with human preferences using both offline and online-generated pairwise preference data. Furthermore, we extended PSO to style transfer and concept customization tasks by constructing pairwise source-target image pairs. Our approach provides an efficient solution for adapting timestep-distilled diffusion models to downstream applications without the need for costly re-distillation. Our work can be further strengthened with enhanced fine-tuning results in personalized generation, and including experiments in controllable image generation.

**Limitations.** While our method achieves initial promising results in style transfer and concept customization, further explorations on loss design and regularization are required to achieve more competitive results with enhanced image quality.

## ETHICAL STATEMENT

Our research on fine-tuning timestep-distilled diffusion models with the proposed PSO generally does not have ethical concerns, except the usage of human preference data. We acknowledge the potential for introducing or amplifying biases present in human judgments, which could lead to unfair or discriminatory outputs. To address this, we commit to implementing diverse sampling strategies for both annotators and image datasets, and developing fairness metrics to assess and mitigate potential biases.

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

## APPENDIX

## A DERIVATIONS

### A.1 PSO DERIVATION

In this section, we present the derivation for Eq. 3 in Sec. 2.1. Given the forward sampled data trajectory $x_{t_0:t_N}^\tau | c$ and reverse sampled reference trajectory $x_{t_0:t_N}^\rho | c$, we represent our PSO objective as increasing the likelihood gap between two trajectories,

$$\mathcal{L} = -\mathbb{E}_{(x_{t_0:t_N}^\tau, x_{t_0:t_N}^\rho, c)} \left[ \log \sigma \left( \beta \log \frac{p_\theta(x_{t_0:t_N}^\tau | c)}{p_{\text{pre}}(x_{t_0:t_N}^\tau | c)} - \beta \log \frac{p_\theta(x_{t_0:t_N}^\rho | c)}{p_{\text{pre}}(x_{t_0:t_N}^\rho | c)} \right) \right], \quad (8)$$

With the formulation in Eq. 2, we can write the joint likelihood in the above equation as,

$$\mathcal{L} = -\mathbb{E}_{(x_{t_0:t_N}^\tau \sim p_{data}(x_{t_0:t_N}^\tau), x_{t_0:t_N}^\rho \sim p_\theta(x_{t_0:t_N}^\rho | c))} \left[ \log \sigma \left( \right. \right.$$
$$\left. \left. \beta \sum_{n=1}^N \left( \log \frac{p_\theta(x_{t_{n-1}}^\tau | x_{t_n}^\tau, c)}{p_{\text{pre}}(x_{t_{n-1}}^\tau | x_{t_n}^\tau, c)} - \log \frac{p_\theta(x_{t_{n-1}}^\rho | x_{t_n}^\rho, c)}{p_{\text{pre}}(x_{t_{n-1}}^\rho | x_{t_n}^\rho, c)} \right) \right) \right]. \quad (9)$$

For the first log-likelihood ratio in the bracket, it can be further derived as,

$$L_1 = -\mathbb{E}_{x_{t_0:t_N}^\tau \sim p_{data}(x_{t_0:t_N}^\tau)} \left[ \log \sigma (\beta \sum_{n=1}^N \log p_\theta(x_{t_{n-1}}^\tau | x_{t_n}^\tau, c) - \beta \sum_{n=1}^N \log p_{\text{pre}}(x_{t_{n-1}}^\tau | x_{t_n}^\tau, c)) \right]$$

$$= -\mathbb{E}_{x_{t_0:t_N}^\tau \sim p_{data}(x_{t_0}^\tau) q(x_{t_1:t_N} | x_{t_0})}$$
$$\left[ \log \sigma (\beta \sum_{n=1}^N \log p_\theta(x_{t_{n-1}}^\tau | x_{t_n}^\tau, c) - \beta \sum_{n=1}^N \log p_{\text{pre}}(x_{t_{n-1}}^\tau | x_{t_n}^\tau, c)) \right]$$

$$(10)$$

Given that each sampling step in the forward process can be written as $x_{t_n} \sim q(x_{t_n}^\tau | x_{t_{n-1}}^\tau, x_0^\tau)$ and we know the marginal $q(x_{t_n} | x_0)$, we can sample $x_{t_{n-1}}^\tau \sim q(x_{t_{n-1}}^\tau | x_{t_n}^\tau, x_0)$ given $x_{t_n}^\tau, x_0^\tau$ with the Bayes equation. In this way, the sub-objective above can be further derived as,

$$L_1 = -\mathbb{E}_q \left[ \log \sigma (\beta \sum_{n=1}^N \log \frac{p_\theta(x_{t_{n-1}}^\tau | x_{t_n}^\tau, c)}{q(x_{t_{n-1}}^\tau | x_{t_n}^\tau, x_0^\tau)} - \beta \sum_{n=1}^N \log \frac{p_{\text{pre}}(x_{t_{n-1}}^\tau | x_{t_n}^\tau, c)}{q(x_{t_{n-1}}^\tau | x_{t_n}^\tau, x_0^\tau)}) \right]$$

$$= -\mathbb{E}_q \left[ \log \sigma \left( \beta (D_{\text{KL}}[q(x_{t_{n-1}}^\tau | x_{t_n,t_0}^\tau) \| p_{\text{pre}}(x_{t_{n-1}}^\tau | x_{t_n}^\tau, c)] - D_{\text{KL}}[q(x_{t_{n-1}}^\tau | x_{t_n,t_0}^\tau) \| p_\theta(x_{t_{n-1}}^\tau | x_{t_n}^\tau, c)]) \right) \right]$$

$$= -\mathbb{E}_q \left[ \log \sigma \left( -\beta \omega_{t_n} (\|\epsilon^\tau - \epsilon_\theta(x_{t_n}^\tau, t_n, c)\|^2 - \|\epsilon^\tau - \epsilon_{pre}(x_{t_n}^\tau, t_n, c)\|^2) \right) \right], \quad (11)$$

where the third equation adopts the reparameterization in the original diffusion objective derivation (Black et al., 2023), and $\omega_{t_n}$ is the reweighting which is usually set to constant in practice.

Meanwhile, the second log-likelihood ratio in Eq. 9, can be further simplified with Eq. 2 as,

$$\log \frac{p_\theta(x_{t_{n-1}}^\rho | x_{t_n}^\rho, c)}{p_{\text{pre}}(x_{t_{n-1}}^\rho | x_{t_n}^\rho, x_0^\rho)} = -\frac{1}{2\sigma_{t_n}^2} (\|x_{t_{n-1}}^\rho - \mu_\theta(x_{t_n}^\rho, t_n, c)\|_2^2 - \|x_{t_{n-1}}^\rho - \mu_{\text{pre}}(x_{t_n}^\rho, t_n, c)\|_2^2) \quad (12)$$

By substituting Eq. 11 and Eq. 12 into Eq. 9 and removing the last sampling step $t_1$, we got the PSO objective as in Eq. 5

$$\mathcal{L}_{\text{PSO}} = -\mathbb{E} \left[ \log \sigma \left( -\beta \cdot \sum_{n=2}^N \left( (\|\epsilon^\tau - \epsilon_\theta(x_{t_n}^\tau, t_n, c)\|^2 - \|\epsilon^\tau - \epsilon_{pre}(x_{t_n}^\tau, t_n, c)\|^2) \right. \right. \right.$$
$$\left. \left. \left. -\frac{1}{2\sigma_{t_n}^2} \left( \|x_{t_{n-1}}^\rho - \mu_\theta(x_n^\rho, t_n, c)\|^2 - \|x_{t_{n-1}}^\rho - \mu_{pre}(x_n^\rho, t_n, c)\|^2 \right) \right) \right) \right] \quad (13)$$

---

**Algorithm 1** Pairwise Sample Optimization (PSO)

---

**Require:** Pre-trained timestep-distilled model $\theta_{\text{pre}}$, prompt $c$, number of timesteps $N$
**Require:** Learning rate $\eta$, regularization weight $\beta$
 1: **while** not converged **do**
 2:    Sample $x_0^\tau \sim p_{\text{data}}(\cdot|c)$               ▷ Data sample
 3:    Sample $x_0^\rho \sim p_\theta(\cdot|c)$               ▷ Reference sample
 4:    Get forward trajectory $\{x_{t_n}^\tau\}_{n=1}^N$ via $q(x_{t_n}^\tau|x_{t_{n-1}}^\tau)$
 5:    Get reverse trajectory $\{x_{t_n}^\rho\}_{n=1}^N$ via $p_\theta(x_{t_{n-1}}^\rho|x_{t_n}^\rho, c)$
 6:    Update $\theta$ using loss $\mathcal{L}_{\text{PSO}}$ in Equation 3
 7: **end while**
**Ensure:** Fine-tuned timestep-distilled model $\theta$

---

**Algorithm 2** Online Pairwise Sample Optimization (Online-PSO)

---

**Require:** Pre-trained timestep-distilled model $\theta_{\text{pre}}$, prompt $c$, number of timesteps $N$
**Require:** Learning rate $\eta$, regularization weight $\beta$, reward model $R(\cdot)$
 1: **while** not converged **do**
 2:    Sample two trajectories $\{x_{t_n}^1\}_{n=1}^N$, $\{x_{t_n}^2\}_{n=1}^N$ via $p_\theta(\cdot|c)$
 3:    Score trajectories: $s_1 = R(x_0^1)$, $s_2 = R(x_0^2)$
 4:    Assign $\{x_{t_n}^\tau\} = \{x_{t_n}^1\}$, $\{x_{t_n}^\rho\} = \{x_{t_n}^2\}$ if $s_1 > s_2$
 5:    Otherwise, $\{x_{t_n}^\tau\} = \{x_{t_n}^2\}$, $\{x_{t_n}^\rho\} = \{x_{t_n}^1\}$
 6:    Update $\theta$ using loss $\mathcal{L}_{\text{PSO-Online}}$ in Equation 7
 7: **end while**
**Ensure:** Fine-tuned timestep-distilled model $\theta$

---

### A.2 MDP FORMULATION FOR SDXL-TURBO

The MDP formulation presented in Sec. 2.1 is applicable to DMD (Yin et al., 2024a) and LCM (Luo et al., 2023). Here we present the MDP formulation for SDXL-Turbo (Sauer et al., 2023). The Euler ancestral sampler can be defined as,

$$x_{\sigma_{n-1}} = x_{\sigma_n} + s_\theta(x_{\sigma_n}; \sigma_n) \cdot (\bar{\sigma}_n - \sigma_n) + \hat{\sigma}_n \cdot z, \ \ z \sim N(0, I), \tag{14}$$

where $\bar{\sigma}_n = \sqrt{(\sigma_n^2 - \sigma_{n-1}^2)\frac{\sigma_{n-1}^2}{\sigma_n^2}}, \hat{\sigma}_n = \sqrt{\sigma_{n-1}^2 - \sigma_n^2}, s_\theta(x, \sigma_n) = (x_{\sigma_n} - f_\theta(x_{\sigma_n}, \sigma_n, c))/\sigma_n$, and $\sigma_n$ is the designated noise levels in Euler ancestral sampler used in SDXL-Turbo which has one-to-one mapping to $t_n$.

The MDP formulation can then be represented as, The Markov Decision Process for distilled models can be then formulated as,

$$s_n = (x_{\sigma_n}, \sigma_n), \ \ a_n = x_{\sigma_{n-1}}, \ \ P(s_{n+1}|s_n, a_n) = \delta(x_{\sigma_{n-1}}, \sigma_{n-1}, c)$$
$$\pi_\theta(a_n|s_n) = N(x_{\sigma_n} + s_\theta(x_{\sigma_n}, \sigma_n)(\bar{\sigma}_n - \sigma_n), \hat{\sigma}_n^2 I),$$

## B SUPPLEMENTARY EXPERIMENTAL SETTINGS AND RESULTS

### B.1 HUMAN PREFERENCE TUNING

**Datasets.** We adopt the Pick-a-Pic v2 (Kirstain et al., 2023) dataset for the offline human preference tuning task following Wallace et al. (2024). It consists of pairwise preferences for images generated by SDXL-beta and Dreamlike, a fine-tuned version of SD1.5 (Rombach et al., 2022). The prompts and preferences were collected from users of the Pick-a-Pic web application. After removing $\sim 12\%$ pairs with tied preference, we obtain $\sim 850K$ win-lose pairs, which are used as data-reference pairs within our Offline-PSO objective as in Eq. 6. For the online human preference setting, we use a subset of 4K prompts from Pick-a-Pic training prompts as the training prompts, and we sample a pair of images given a prompt, assign the target & reference labels based on the PickScore (Kirstain et al., 2023) as a discriminator.

---

**Algorithm 3** Offline Pairwise Sample Optimization (Offline-PSO)

---

**Require:** Pre-trained timestep-distilled model $\theta_{\text{pre}}$, prompt $c$, number of timesteps $N$
**Require:** Learning rate $\eta$, regularization weight $\beta$, pre-sampled reference set $\mathcal{R}$
 1: **while** not converged **do**
 2:      Sample $x_0^{\tau} \sim p_{\text{data}}(\cdot|c)$                              ▷ Data sample
 3:      Sample $x_0^{\rho} \sim \mathcal{R}$                                  ▷ Reference sample
 4:      Get forward trajectories $\{x_{t_n}^{\tau}\}_{n=1}^{N}$, $\{x_{t_n}^{\rho}\}_{n=1}^{N}$ via $q(\cdot|\cdot)$
 5:      Update $\theta$ using loss $\mathcal{L}_{\text{PSO-Offline}}$ in Equation 6
 6: **end while**
**Ensure:** Fine-tuned timestep-distilled model $\theta$

---

**Parameter Efficient Fine-tuning.** Diffusion models have been repurposed for various tasks with effective tuning methods (He et al., 2023; He & Aliaga, 2024; Song et al., 2024a; He & Aliaga, 2023; Ruiz et al., 2023). As we aim to design a lightweight method for fine-tuning distilled diffusion models, it's natural for us to adopt the parameter-efficient fine-tuning methods which majorly are based on matrix low-rank decomposition. LoRA (Hu et al., 2021) and following works (Liu et al., 2024b; Dettmers et al., 2024) decompose the weight matrix along the channel dimension into two low-rank matrices, which induces much lower costs in fine-tuning compared with tuning the original full-rank weight. Meanwhile, (Chen et al., 2024b) proposes the filter subspace decomposition for weight matrices. The filter subspace decomposition method (Qiu et al., 2018) has shown effectiveness in continual learning (Miao et al., 2021a; Chen et al., 2024a), video representation learning (Miao et al., 2021b), graph learning (Cheng et al., 2021), and generative tasks (Wang et al., 2021a; 2019; 2021b; Li et al., 2024). There are some other works on fine-tuning the SVD decomposition of weights(Han et al., 2023), Kronecker decomposition (Patel et al., 2024), sparsity (Wang et al., 2024), non-linearity (Zhong et al., 2024) or fine-tunig bias parameters(Xie et al., 2023). Considering the implementation and the integration with other codebases, we choose LoRA (Hu et al., 2021) to fine-tune timestep-distilled diffusion models.

**Training Details.** We use LoRA (Hu et al., 2021) to fine-tune all the distilled diffusion models efficiently, we set LoRA rank $r = 16$ for SDXL-DMD2 and SDXL-Turbo, and $r = 32$ for SDXL-LCM in both online and offline human preference tuning experiments. As for other training hyperparameters, we set the number of training distilled steps $N = 4$, which is the same as the number of sampling steps of these distilled models, and we set the regularization weight $\beta = 50$ for Offline preference tuning, and $\beta = 5$ for Online preference tuning. We set the batch size to 64 and the learning rate to $1e-5$ for offline experiments, and train for 5k steps. For online preference tuning, we first sample 128 pairs of images with 128 training prompts, which are further labeled as reference and target with PickScore (Kirstain et al., 2023) with a batch size of 64, and then we conduct online PSO on the sampled pairs for 1 epoch with a batch size of 32 and learning rate $1e-5$ and train for 20k steps.

Table 3: Results of SDXL-LCM

| Dataset | Method | PickScore | CLIP Score | ImageReward | Aesthetic Score |
|---------|--------|-----------|------------|-------------|-----------------|
| Pickapic Test | SDXL-LCM | 21.9072 | 0.3572 | 0.6717 | 5.8642 |
| | SDXL + LCM-LoRA | 21.8520 | 0.3553 | 0.6840 | 5.9447 |
| | SDXL-DPO + LCM-LoRA | 22.3178 | 0.3633 | 0.7883 | 5.9286 |
| | Offline-PSO w/ SDXL-LCM | 22.3953 | **0.3668** | 0.8129 | 5.9949 |
| | Online-PSO w/ SDXL-LCM | **22.6353** | 0.3613 | **0.8343** | **6.0117** |
| Parti-Prompts | SDXL-LCM | 22.4761 | 0.3492 | 0.6738 | 5.5867 |
| | SDXL + LCM-LoRA | 22.4332 | 0.3481 | 0.6571 | 5.5141 |
| | SDXL-DPO + LCM-LoRA | 22.7854 | 0.3551 | 0.7778 | 5.7090 |
| | Offline-PSO w/ SDXL-LCM | 22.8131 | **0.3592** | 0.8131 | 5.8367 |
| | Online-PSO w/ SDXL-LCM | **22.9451** | 0.3581 | **0.8351** | **5.8572** |

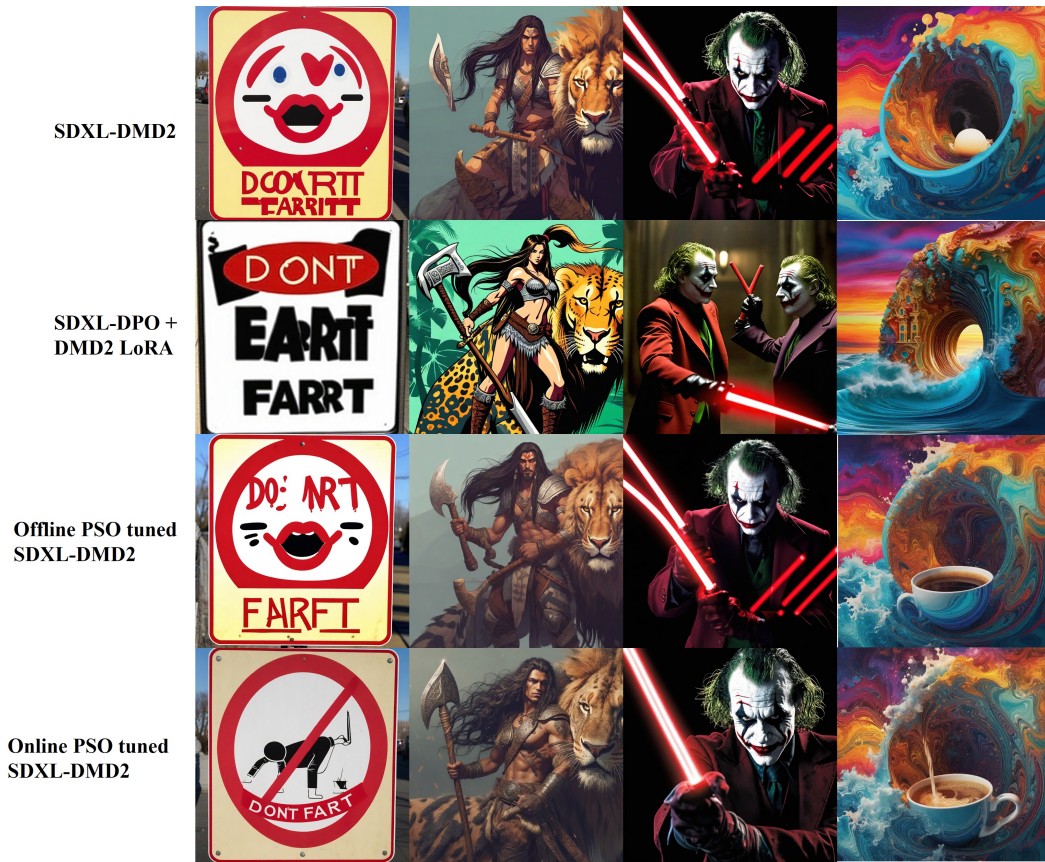

Figure 6: Human Preference Tuning with SDXL-DMD2 additional results, Prompts from left to right: Sign that says DON'T FART// Warrior with an axe on a exotic animal with long hair style loisel// Movie Still of The Joker wielding a red Lightsaber, Darth Joker a sinister evil clown prince of crime, HD Photograph// A swirling, multicolored portal emerges from the depths of an ocean of coffee, with waves of the rich liquid gently rippling outward. The portal engulfs a coffee cup, which serves as a gateway to a fantastical dimension. The surrounding digital art landscape reflects the colors of the portal, creating an alluring scene of endless possibilities.

**Evaluation.**   For all baseline distilled models and our PSO fine-tuned models, we conduct testing on Pickapic (Kirstain et al., 2023) test and PartiPrompts (Yu et al., 2022). They contain 500 and 1632 prompts respectively. We sample for each prompt a testing image with 4 sampling steps (we also conduct a 1-step sampling test for SDXL-Turbo, which is trained under a 4-step schedule). Each sampled image is then evaluated by PickScore (Kirstain et al., 2023), CLIP Score (Radford et al., 2021), ImageReward (Xu et al., 2024), and Aesthetic Score (Schuhmann et al., 2022). We report the average scores in the tables.

**Additional Results.**   We provide more qualitative results for SDXL-DMD2 human preference tuning in Figure 6 & 7. As shown in those figures, images generated by our Online-PSO and Offline-PSO tuned models demonstrate better aesthetic values, prompt following, and detailed accuracy. For SDXL-Turbo, we provide qualitative illustrations in Figure 8a for 1-step evaluation and Figure 8b for 4-step evaluation. Again, we see that our PSO-tuned models generate more visual-appeal images in both 1-step and 4-step generations. Specifically, we observe better visual detail generation, for instance, human body, objects, and object consistency in images generated by our method. And finally, we provide quantitative results for SDXL-LCM (Luo et al., 2023) in Table 3. As shown in the table, our method achieves higher results in all metrics compared with baselines. Specifically, our method achieves better results compared with SDXL-DPO + LCM-LoRA, which is the preference-tuned multi-step SDXL equipped with LCM LoRA, which further justify and validate our method.

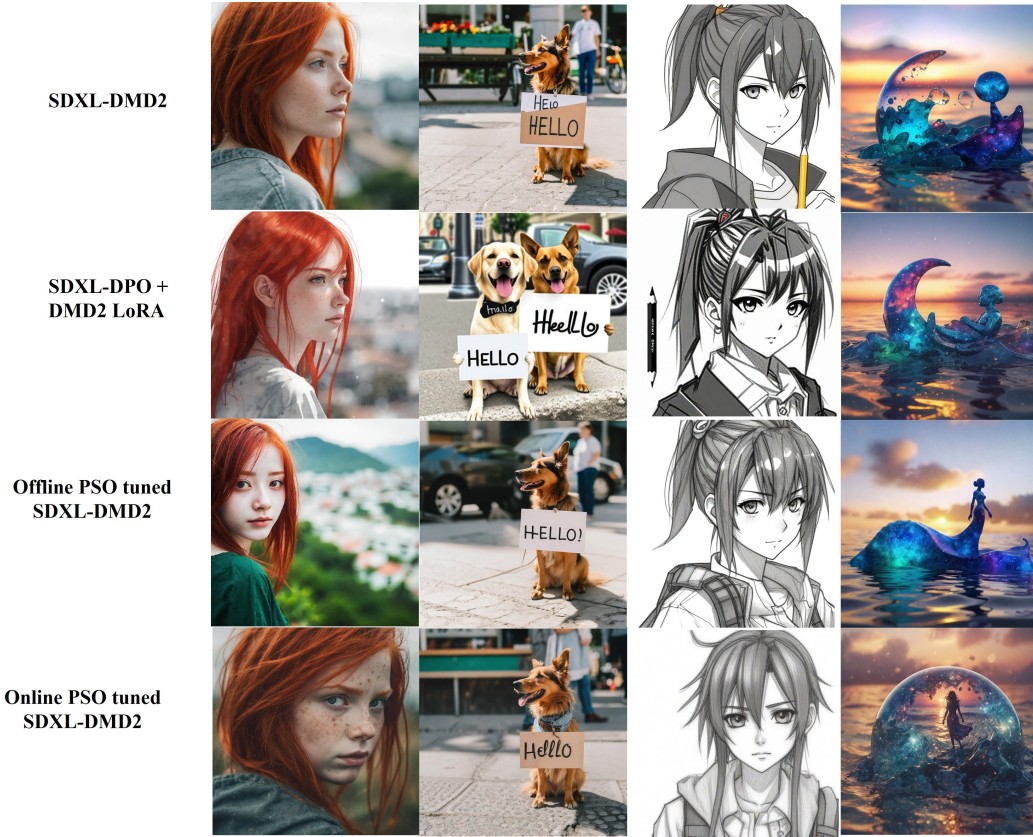

Figure 7: Human Preference Tuning with SDXL-DMD2 additional results. Prompts from left to right: a red hair girl look at you, distant view.// A dog with a sign that reads "Hello".// anime pencil concept style.// A galaxy-colored figurine floating over the sea at sunset, photorealistic.

Table 4: Quantitative evaluation of PSO fine-tuning on concept customization task.

| Model | Inference Steps | CLIP-T | DINO | CLIP-I |
|---|---|---|---|---|
| Dreambooth w/ SDXL | 50 | 0.256 | **0.610** | **0.741** |
| Dreambooth w/ SDXL-Turbo | **4** | 0.267 | 0.151 | 0.326 |
| (Ours) PSO w/ SDXL-Turbo | **4** | **0.271** | 0.593 | 0.733 |

## B.2 PSO FOR STYLE TRANSFER

We adopt SDXL-Turbo (Sauer et al., 2023) for this experiment with the Pokemon dataset (Pinkney, 2022). The reference images are sampled using the training prompts in the pokemon dataset. We add LoRA of rank $r = 16$ to the attention layers and set $\beta = 50$. Then, we adopt the objective of PSO for fine-tuning. We use the batch size of 8, learning rate of 1e-5, and train for 3k steps.

## B.3 PSO FOR CONCEPT CUSTOMIZATION

We also use SDXL-Turbo for this task with the Dreambooth dataset (Ruiz et al., 2023). We adopt the Offline-PSO objective in Eq. 6 in the way that we use the given input images as the targeted set, and sample reference images from SDXL-Turbo with the prompt 'A photo of [V] concept'. As mentioned previously, we further train the model with a variant of the Eq. 6 where we remove the reference model $\epsilon_{\text{pre}}$ and add prior preservation loss with images sampled from the initial model following Ruiz et al. (2023); Qiu et al. (2023). We set lora rank $r = 16$, $\beta = 5$, learning rate of 5e-5 and train for 1000 steps with batch size of 4.

We provide additional results in Figure 9. Our PSO can effectively customize the distilled model with the given animal or object.

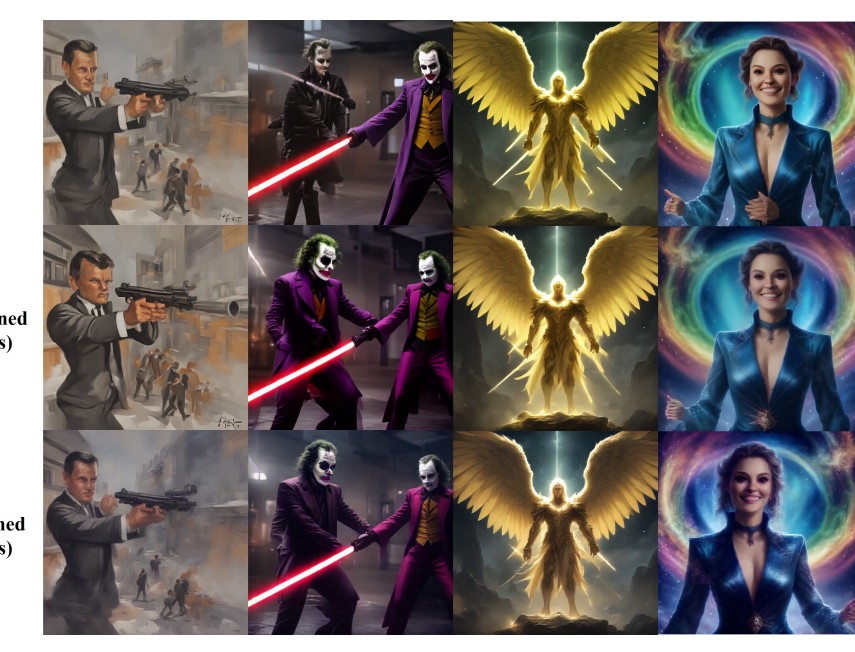

(a) Preference Fine-tuning Results with PSO on 1step SDXL-Turbo.

(b) Preference Fine-tuning Results with PSO on 4step SDXL-Turbo

Figure 8: Experiments of Tuning SDXL-Turbo for human preference tuning.

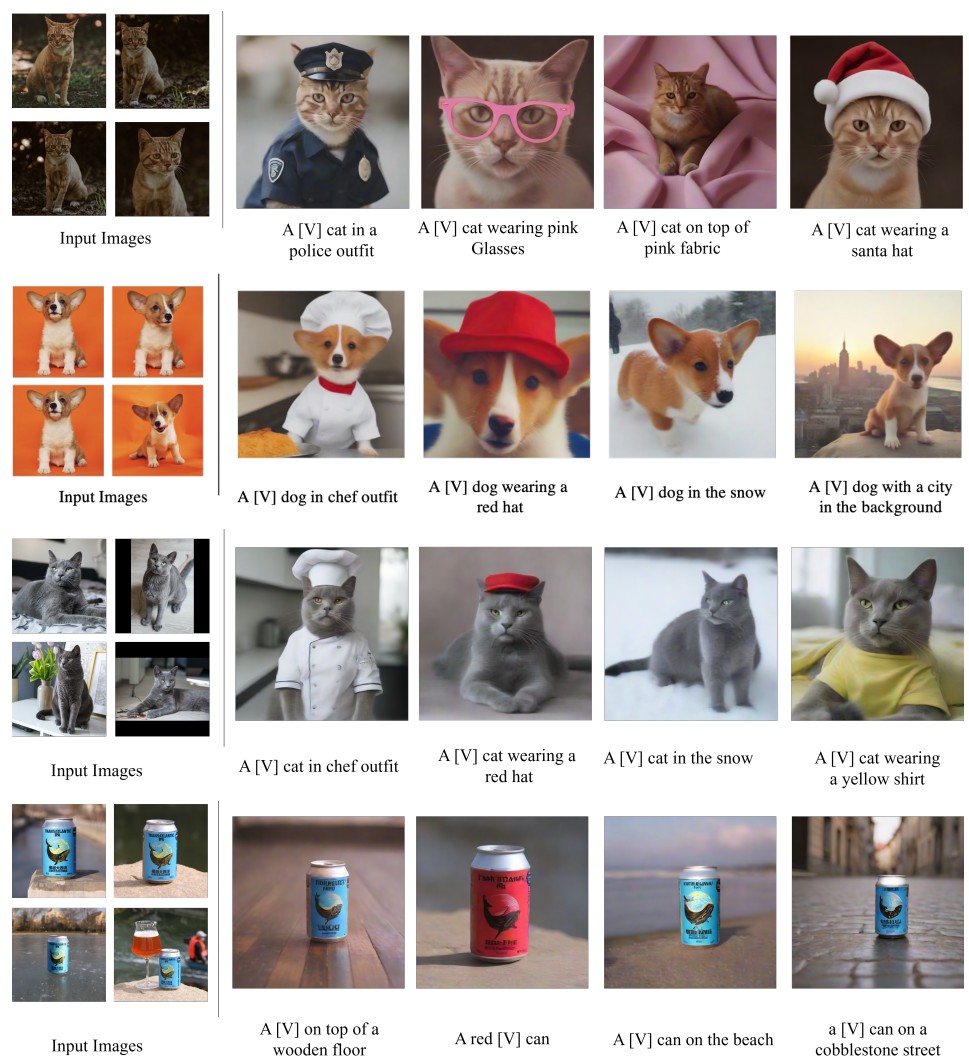

Figure 9: Experiments of Tuning SDXL-Turbo for personalized generation with PSO. The proposed method can effectively tune SDXL-Turbo to generate images that contain the given objects.

