# OpenReview forum: "Tuning Timestep-Distilled Diffusion Model Using Pairwise Sample Optimization"
_ICLR.cc/2025/Conference — ICLR 2025 Poster_

### Official Review · Reviewer_bSbN · 2024-10-23

**Soundness:** 3
**Presentation:** 2
**Contribution:** 3
**Rating:** 6
**Confidence:** 4

**Summary:**

This paper proposes PSO, an objective for finetuning timestep-distilled diffusion models like SDXL-DMD2, SDXL-Turbo, and SDXL-LCM. These models, trained with GAN loss or consistency distillation loss, can no longer be trained with traditional diffusion loss. The proposed PSO is a modification of DPO, which is widely used in human preference optimization for LLMs and diffusion models. Unlike DPO, PSO uses negative samples generated by timestep-distilled models and supports both offline and online generation. PSO shows effectiveness in human preference tuning, style transfer, and concept customization for these models.

**Strengths:**

- This paper addresses the important problem of finetuning timestep-distilled diffusion models.

- PSO, while similar to DPO, introduces key differences, notably the ability to support both offline and online training.

- Finetuning timestep-distilled models with PSO proves more effective than attaching timestep-distilled LoRA to finetuned diffusion models, and the experiments demonstrate the potential for style transfer and customization.

**Weaknesses:**

- The paper primarily demonstrates the effectiveness of PSO in human preference tuning. However, direct finetuning of timestep-distilled models holds more significance in concept customization rather than human preference tuning. As the authors mention in the introduction, customization methods like Dreambooth require significantly less time compared to timestep distillation methods like DMD. On the other hand, human preference tuning, such as DPO, still requires lengthy training and may not offer much advantage over timestep distillation. To further highlight the necessity of PSO in human preference tuning, it would be beneficial to compare the computational cost of timestep distillation versus human preference tuning.

- For the reasons mentioned above, PSO should further showcase its capabilities in concept customization and style transfer on datasets from Dreambooth (Ruiz et al.) and StyleDrop (Sohn et al). Specifically, following these works, text-image alignment could be compared using CLIP, and image-reference similarity could be assessed using DINO. Additionally, results in Figures 5 and 9 appear blurry, so a quantitative comparison using CLIP and DINO is essential.

- PSO uses generated negative samples, which is similar to SPIN [A]. A discussion on the differences between PSO and SPIN would be beneficial. A discussion on the differences between PSO and SPIN could be added to the related works section.

[A] Chen et al., Self-Play Fine-Tuning Converts Weak Language Models to Strong Language Models, ICML 2024.

**Questions:**

- Why was offline PSO used for style transfer and online PSO for concept customization? Could the reverse approach also work?

- In Tables 1 and 2, is there a specific reason why "Offline-DPO" and "Online-DPO" are listed instead of "Offline-PSO" and "Online-PSO"?

- What reward model was used for online PSO?

---

> ### Author Response · Authors · 2024-11-25
> **Response to Reviewer bSbN**
>
> Thanks to the reviewer for the constructive feedback, we provide our responses below and hope it can alleviate your concerns.
>
> **Q1: Compare the computational costs of distillation with human preference tuning using PSO.**
>
> We compare the computational cost of the distillation of SDXL-DMD2 with PSO tuning for human preference by measuring their training GPU hours. Note that a similar level of computational cost of human preference tuning is also reported in MaPO[1] (our cost is lower than MaPO as we remove the redundant data from the dataset). As shown in the table below, the computational cost of human preference tuning with Offline-PSO is significantly less than the diffusion model distillation. Specifically, the preference tuning cost of PSO is roughly **2.0%(78/3840)** of the distillation cost of DMD2. Thus, our method demonstrates significant advantages in training efficiency compared with tuning followed by timestep distillation.
>
> | Method     | SDXL-DMD2 Distillation | SDXL-LCM Distillation | **PSO w/ SDXL-DMD2** | **PSO w/ SDXL-LCM** |
> |------------|-------------------|------------------|----------------------|---------------------|
> | A100 Hours | 3840              | 1280             | **78**               | **78**              |
>
>
>
> **Q2: More quantitative results using CLIP and DINO scores.**
>
> Thanks to the reviewer for bringing this out. We conduct quantitative experiments with the proposed method on concept customization. Specifically, we adopt the experimental setup in Dreambooth by evaluating our method and the baseline with all the 30 concepts in the Dreambooth dataset. For evaluation, we sample images with 25 test prompts for each concept. As for the metrics, we follow the reviewer's suggestion and use the CLIP score for text-image alignment (**CLIP-T**), and the CLIP score with DINO score for fidelity (**DINO** and **CLIP-I**). We compare our method with customizing the multi-step diffusion model (Dreambooth w/ SDXL) and the distilled baseline model before customization (SDXL-Turbo).
>
>
> | Model                       | Inference Steps | CLIP-T     | DINO       | CLIP-I     |
> |-----------------------------|-----------------|------------|------------|------------|
> | Dreambooth w/ SDXL          | 50              | 0.256     | **0.610** | **0.741** |
> | SDXL-Turbo                   | **4**           | 0.267     | 0.131     | 0.216     |
> | **(Ours) PSO w/ SDXL-Turbo** | **4**           | **0.271** | 0.593     | 0.733   |
>
> As shown in the table above, the proposed PSO demonstrates effectiveness given the quantitative results. Our method shows superior results than SDXL-Turbo and comparable results with multi-step customization baseline. Specifically, compared with Dreambooth w/ SDXL, our method achieves a higher text-alignment CLIP-T score while using a magnitude fewer inference steps.
>
>
> **Q3: Comparison with SPIN.**
>
> Thanks to the reviewer for bringing this out. Our method is targeting at tuning few-step timestep-distilled diffusion models, whereas SPIN fine-tunes the multi-step regular diffusion models. We also show the effectiveness of PSO in general diffusion model tuning tasks including preference tuning, style transfer, and concept customization, while SPIN solely focuses on preference tuning. Moreover, we would like to state that SPIN consists of training iterations, where each iteration can be seen as Offline-PSO with offline sampled reference data, thus SPIN can be viewed as a multi-stage Offline-PSO. We add this discussion in the related works.
>
>
> **Q4: Why offline PSO for style transfer and online for customization.**
>
> We would like to clarify that we do not use Online-PSO for customization where both target and reference images are sampled from the distilled diffusion model. As stated in Line 459, we adopt the PSO objective for customization defined in Eq. (3), where the reference images are sampled throughout the training. We adopt the Offline-PSO setting for style transfer for efficiency concerns as it takes a longer training time. We conduct an ablation style transfer experiment with Eq. (3) and it works as well as Offline-PSO as in the paper.
>
>
> **Q5: Table typos.**
>
> Thanks to the reviewer for pointing out the typos. We have changed that in the revision.
>
> **Q6: Reward model for online PSO.**
>
> We adopt Pickscore in the Online-PSO for human preference tuning. We will clarify that in the revision.
>
> **Reference:**
>
> 1. Hong, J., Paul, S., Lee, N., Rasul, K., Thorne, J., & Jeong, J. (2024). Margin-aware Preference Optimization for Aligning Diffusion Models without Reference. arXiv preprint arXiv:2406.06424.

---

> > ### Comment · Reviewer_bSbN · 2024-11-25
> >
> > The authors' rebuttal addresses my concerns. I will raise my score to 6. Regarding the DINO and CLIP scores in the rebuttal, I think it would be beneficial to also compare the DINO and CLIP scores obtained by loading the LoRA trained on SDXL into LCM, following the LCM-LoRA [1].
> >
> > [1] Luo, Simian, et al. "Lcm-lora: A universal stable-diffusion acceleration module."

---

> > > ### Author Response · Authors · 2024-11-26
> > > **Thank you for the positive feedback**
> > >
> > > Thanks to the reviewer for the positive feedback. We greatly appreciate your efforts and time in the review process. We will include all the comparisons in the final revision.

---

### Official Review · Reviewer_i4cT · 2024-10-31

**Soundness:** 4
**Presentation:** 3
**Contribution:** 4
**Rating:** 8
**Confidence:** 4

**Summary:**

The motivation behind this work stems from the observation that models distilled over multiple timesteps can suffer from blurriness and reduced quality if distilled further using a naive diffusion objective. A straightforward solution would be to use a teacher model and repeat the distillation process, but this approach is overly cumbersome.
To address this, the authors propose an optimization method that fine-tunes the distilled model without requiring a teacher model. PSO introduces additional self-generated images and optimizes the relative likelihood margin, preserving the model's ability to generate high-quality images in fewer steps while still allowing fine-tuning.

**Strengths:**

- This problem is both novel and relevant, as many time-distilled diffusion models incorporate multiple objectives, such as adversarial training, making efficient fine-tuning essential.

- The proposed method’s use of pair optimization is innovative, providing a critical solution to eliminate the need for the cumbersome teacher model distillation process.

- Extensive experiments underscore the effectiveness of this approach.

**Weaknesses:**

- The notation used in the paper lacks clarity and can be confusing for readers, particularly with symbols like ρ and p.

- Additionally, Figure 2 is somewhat abstract. Including an algorithm for the complete method would improve readability and help readers understand the approach more efficiently.

- it would be beneficial to discuss how this method compares to current DiT-like image generation models, such as Flux.

**Questions:**

This paper is good to me. I hope author can provide an algorithm for complete method for better readability.

---

> ### Author Response · Authors · 2024-11-25
> **Response to Reviewer i4cT**
>
> Thanks to the reviewer for acknowledging the novelty and our contribution. We respond to your questions below.
>
> **Q1: Notation clarification.**
>
> Thanks to the reviewer for pointing this out. We will change the notation of the reference image $\rho$ in the revision.
>
>
> **Q2: Additional algorithm table.**
>
> Thanks to the reviewer for bringing this out. We provide an algorithm table in the revision.
>
>
> **Q3: DiT based diffusion models.**
>
> Thanks to the reviewer for this question. Our method is agnostic to model architectures. While we mainly validate our methods on U-Net based timestep-distilled diffusion models, i.e., SDXL distillation variants, we suppose the method can also be applied to DiT-based distilled models like FLUX-schnell. We will explore it in the future work.

---

### Official Review · Reviewer_fXQz · 2024-11-01

**Soundness:** 2
**Presentation:** 2
**Contribution:** 2
**Rating:** 6
**Confidence:** 4

**Summary:**

This paper focuses on fine-tuning distilled diffusion models. The authors introduce a pairwise optimization approach, which combines preference optimization over positive-negative pairs and alignment with a frozen distilled few-step diffusion model. They present both offline and online training versions of this loss function. Experiments on tasks of human-preference tuning, style transfer, and concept customization show it outperforms the original DPO.

**Strengths:**

Fine-tuning time-distilled diffusion models is very useful.

The proposed method generalizes well across different tasks and base models.

**Weaknesses:**

Although the experiments cover many tasks and base models, there aren’t many comparisons with other methods. The authors only compare their method to the original DPO in Table 1 and Table 3, and there are no comparisons in Table 2. Including comparisons with more fine-tuning methods would make the results more comprehensive and highlight the advantages of the proposed approach.

**Questions:**

Why is DPO used in all the tables instead of your method, PSO?

In Table 1, why are your results bolded even when they perform worse than SDXL-DPO?

How should we interpret that the online update version shows improvements in almost all metrics but generally performs worse than the offline version on the CLIP score?

---

> ### Author Response · Authors · 2024-11-25
> **Response to Reviewer fxQz (Part 1/2)**
>
> Thanks to the reviewer for the constructive feedback. We provide our response below and hope it can alleviate your concerns.
>
> **Q1: More fine-tuning baselines for comparisons.**
>
> Thanks to the reviewer for bringing this up. We add more baselines including the supervised fine-tuning(SDXL-SFT), RPO[1], MaPO[2], and SPO[3] as additional human preference tuning baselines in Table 1 in the revision and show it below. Note that the proposed PSO targets directly tuning distilled diffusion models in a **few-step(1-4 step)** generation regime, while other fine-tuning baselines including DPO tune the **multi-step(50 steps)** diffusion models. To make it a fair comparison, we mainly focus on comparing with the few-step distilled models in the tables, indicated by the gray and black text color in Table 1. Nevertheless, as shown in the table below, the distilled few-step model (SDXL-DMD2) tuned with both Online-PSO consistently achieves the best results in Pickscore (as our major focus is human preference) and comparable results on other metrics compared with competitive multi-step diffusion models which are tuned with various human preference tuning baselines. This further highlights the effectiveness of the proposed method.
>
> | Dataset | Method | **Inference Steps** | PickScore | CLIP Score | ImageReward | Aesthetic Score |
> |---------|---------|----------------|-----------|------------|-------------|-----------------|
> | Pickapic Test | SDXL | 50 | 22.30 | 0.3713 | 0.8556 | 6.060 |
> | | SDXL-DPO | 50 | 22.60 | **0.3787** | **1.0075** | 6.040 |
> | | SDXL-SFT | 50 | 22.25 | 0.3693 | 0.8665 | 5.921 |
> | | SDXL-RPO[1] | 50 | 22.65 | 0.3723 | 0.9623 | 6.012 |
> | | SDXL-SPO[3] | 50 | 22.70 | 0.3527 | 0.9417 | **6.283** |
> | | SDXL-MaPO[2] | 50 | 22.50 | 0.3735 | 0.9481 | 6.170 |
> | | SDXL-DMD2 | **4** | 22.35 | 0.3679 | 0.9363 | 5.937 |
> | | SDXL-DPO + DMD2-LoRA | **4** | 22.20 | 0.3673 | 0.9287 | 5.759 |
> | | **(Ours) Offline-PSO w/ SDXL-DMD2** | **4** | 22.46 | 0.3690 | 0.9381 | 5.994 |
> | | **(Ours) Online-PSO w/ SDXL-DMD2** | **4** | **22.73** | 0.3671 | 0.9773 | 6.077 |
> | Parti-Prompts | SDXL | 50 | 22.77 | 0.3607 | 0.9142 | 5.750 |
> | | SDXL-DPO | 50 | 22.92 | **0.3674** | 1.1180 | 5.795 |
> | | SDXL-SFT | 50 | 22.85 | 0.3610 | 0.8565 | 5.675 |
> | | SDXL-RPO[1] | 50 | 22.98 | 0.3670 | 1.0770 | 5.872 |
> | | SDXL-SPO[3] | 50 | 23.27 | 0.3428 | 1.0668 | **6.083** |
> | | SDXL-MaPO[2] | 50 | 22.81 | 0.3661 | 1.0315 | 5.912 |
> | | SDXL-DMD2 | **4** | 22.99 | 0.3607 | 1.0713 | 5.671 |
> | | SDXL-DPO + DMD2-LoRA | **4** | 22.76 | 0.3644 | 1.0638 | 5.513 |
> | | **(Ours) Offline-PSO w/ SDXL-DMD2** | **4** | 23.07 | 0.3649 | 1.0964 | 5.715 |
> | | **(Ours) Online-PSO w/ SDXL-DMD2** | **4** | **23.29** | 0.3634 | **1.1702** | 5.836 |
>
> **Q2: Table typos.**
>
> Thanks to the reviewer for pointing out the typo. We have changed them to the correct names in the revision.
>
> **Q3: Question on bold results.**
>
> As we stated above, our method tunes the few-step distilled diffusion models and we mainly focus on comparing with few-step diffusion model baselines. We provide an updated table that mainly compares the proposed method with few-step baselines below. As shown in the table, the proposed method achieves the best results and thus we make our results bold.
>
> | Dataset | Method | **Inference Steps** | PickScore | CLIP Score | ImageReward | Aesthetic Score |
> |---------|---------|----------------|-----------|------------|-------------|-----------------|
> | Pickapic Test | SDXL-DMD2 | **4** | 22.35 | 0.3679 | 0.9363 | 5.937 |
> | | SDXL-DPO + DMD2-LoRA | **4** | 22.20 | 0.3673 | 0.9287 | 5.759 |
> | | **(Ours) Offline-PSO w/ SDXL-DMD2** | **4** | 22.46 | **0.3690** | 0.9381 | 5.994 |
> | | **(Ours) Online-PSO w/ SDXL-DMD2** | **4** | **22.73** | 0.3671 | **0.9773** | **6.077** |
> | Parti-Prompts | SDXL-DMD2 | **4** | 22.99 | 0.3607 | 1.0713 | 5.671 |
> | | SDXL-DPO + DMD2-LoRA | **4** | 22.76 | 0.3644 | 1.0638 | 5.513 |
> | | **(Ours) Offline-PSO w/ SDXL-DMD2** | **4** | 23.07 | **0.3649** | 1.0964 | 5.715 |
> | | **(Ours) Online-PSO w/ SDXL-DMD2** | **4** | **23.29** | 0.3634 | **1.1702** | **5.836** |

---

> > ### Author Response · Authors · 2024-11-25
> > **Response to Reviewer fxQz (Part 2/2)**
> >
> > **Q4: Questions on results, why online-PSO is the better in all metrics except CLIP score.**
> >
> > We would like to clarify that we solely use PickScore as the reward in tuning the distilled models for enhancing human preference, as stated in Line 347. Therefore, our primary focus is on comparing performance based on the PickScore metric, where our online-PSO consistently outperforms offline-PSO. While other metrics, such as the CLIP score, exhibit a positive correlation with human preference (PickScore), they could show some degree of fluctuation [1,3]. Moreover, online-PSO achieves a higher CLIP score than offline in directly tuning SDXL-Turbo, which is shown in the 4-th row and the last row in Table 2.
> >
> >
> > **References:**
> >
> > 1. Gu, Y., Wang, Z., Yin, Y., Xie, Y., & Zhou, M. (2024). Diffusion-RPO: Aligning Diffusion Models through Relative Preference Optimization. arXiv preprint arXiv:2406.06382.
> >
> > 2. Hong, J., Paul, S., Lee, N., Rasul, K., Thorne, J., & Jeong, J. (2024). Margin-aware Preference Optimization for Aligning Diffusion Models without Reference. arXiv preprint arXiv:2406.06424.
> >
> > 3. Liang, Z., Yuan, Y., Gu, S., Chen, B., Hang, T., Li, J., & Zheng, L. (2024). Step-aware Preference Optimization: Aligning Preference with Denoising Performance at Each Step. arXiv preprint arXiv:2406.04314.

---

> > > ### Comment · Reviewer_fXQz · 2024-11-26
> > >
> > > It might be hard to say "fluctuation" when the different behavior between CLIP Score and other metrics happens in almost all experiments.

---

> > > > ### Author Response · Authors · 2024-11-28
> > > > **Reponses to Follow-up Questions on Interpretation of CLIP Score Results**
> > > >
> > > > Thank the reviewer for the careful observation. This pattern can stem from: **1. difference of training data in Offline and Online PSO**, and **2. property of PickScore adopted in Online-PSO**.
> > > >
> > > >
> > > >
> > > > 1.   **Training Data Difference.** When tuning the distilled models, **Offline-PSO** uses pre-sampled image pairs in PickaPicv2 dataset, where most of the images are **generated with the multi-step diffusion model (SDXL)**. **Online-PSO**, on the other hand, utilizes the image **directly sampled from the distilled model (e.g., SDXL-DMD2)**. As distilled models utilizes the multi-step model as a teacher, their abilities on text-to-image alignment (CLIP score) are upper-bounded by the teacher model to some extent. To illustrate this CLIP score gap on **PickaPic dataset**, we can compare the CLIP scores between SDXL and SDXL-DMD2/ SDXL-Turbo/ SDXL-LCM on pickapic test set, **0.3713(SDXL)** vs. **0.3679 (SDXL-DMD2)**/ **0.3642 (SDXL-Turbo)**/ **0.3572 (SDXL-LCM)**, as shown in Table 1,2 and 3. In this way, this CLIP score gap of the training data can lead to the observed pattern that better CLIP results are achieved by the models tuned with Offline-PSO compared to Online-PSO.
> > > > 2.   **PickScore Property.**
> > > > As shown in the **Figure 4** of the **PickScore paper [4]**, the **disagreement** between CLIP score and PickScore is clearly illustrated. Thus, as Online-PSO directly optimizes PickScore, it may diverge from CLIP's text-image alignment objective, leading to sub-optimal CLIP score results.
> > > >
> > > >
> > > > Lastly, we would like to re-state that our primary focus is on comparing performance based on the PickScore metric, where our online-PSO consistently outperforms offline-PSO.
> > > >
> > > >
> > > > **References**:
> > > >
> > > > 4.Kirstain, Yuval, et al. "Pick-a-pic: An open dataset of user preferences for text-to-image generation." Advances in Neural Information Processing Systems 36 (2023): 36652-36663.([link](https://arxiv.org/pdf/2305.01569))

---

> > > > > ### Comment · Reviewer_fXQz · 2024-11-28
> > > > >
> > > > > Thank you for your response. Most of my concerns have been resolved. Based on the rebuttal, I would like to increase my suggestion to borderline accept. I strongly suggest you to include the new experiments and discussions in your revised version.

---

> > > > > > ### Author Response · Authors · 2024-11-28
> > > > > >
> > > > > > Thank you for the prompt response and for raising the score! We greatly appreciate your time and efforts in the review and rebuttal process, as well as your valuable feedback which has helped improve the quality and clarity of our work. We will certainly incorporate all the new experiments and discussions in the final revision.

---

> > ### Comment · Reviewer_fXQz · 2024-11-26
> >
> > I wonder if new competitors will work when fine-tuning distilled diffusion models? It would be more useful if you could compare them directly under the same settings.

---

> > > ### Author Response · Authors · 2024-11-28
> > > **Reponses to Follow-up Questions on Baselines**
> > >
> > > Thanks to the reviewer for bringing up further discussions on baseline methods. Following the reviewer's suggestion, we conduct complementary experiments on tuning the distilled diffusion model (SDXL-DMD2) with new competitive methods including RPO[1], MaPO[2], and SPO[3], for human preference. We follow the implementation of these methods as well as our experimental settings for both training and evaluation. We report the results in the table below, as shown in the table, directly tuning with RPO[1] and MaPO[2] leads to **degraded results**. Moreover, while SPO[3] leads to minor improvement compared with the original SDXL-DMD2, it still **falls behind the proposed PSO in all metrics**, especially when compared to our Online-PSO.
> > >
> > > | Dataset | Method | **Inference Steps** | PickScore | CLIP Score | ImageReward | Aesthetic Score |
> > > |---------|---------|----------------|-----------|------------|-------------|-----------------|
> > > | Pickapic Test | SDXL-DMD2 | 4 | 22.35 | 0.3679 | 0.9363 | 5.937 |
> > > | | *RPO* [1] *w/ SDXL-DMD2* | *4* | *22.16* | *0.3634* | *0.9172* | *5.786* |
> > > | | *MaPO* [2] *w/ SDXL-DMD2* | *4* | *21.04* | *0.3386* | *0.5428* | *5.138* |
> > > | | *SPO* [3] *w/ SDXL-DMD2* | *4* | *22.37* | *0.3593* | *0.9288* | *5.945* |
> > > | | **(Ours) Offline-PSO w/ SDXL-DMD2** | 4 | 22.46 | **0.3690** | 0.9381 | 5.994 |
> > > | | **(Ours) Online-PSO w/ SDXL-DMD2** | 4 | **22.73** | 0.3671 | **0.9773** | **6.077** |
> > > | Parti-Prompts | SDXL-DMD2 | 4 | 22.99 | 0.3607 | 1.0713 | 5.671 |
> > > | | *RPO* [1] *w/ SDXL-DMD2* | *4* | *22.89* | *0.3582* | *1.0583* | *5.589* |
> > > | | *MaPO* [2] *w/ SDXL-DMD2* | *4* | *21.63* | *0.3424* | *0.6385* | *5.293* |
> > > | | *SPO* [3] *w/ SDXL-DMD2* | *4* | *23.01* | *0.3423* | *1.0143* | *5.696* |
> > > | | **(Ours) Offline-PSO w/ SDXL-DMD2** | **4** | 23.07 | **0.3649** | 1.0964 | 5.715 |
> > > | | **(Ours) Online-PSO w/ SDXL-DMD2** | **4** | **23.29** | 0.3634 | **1.1702** | **5.836** |

---

### Official Review · Reviewer_B3DX · 2024-11-07

**Soundness:** 3
**Presentation:** 2
**Contribution:** 4
**Rating:** 6
**Confidence:** 3

**Summary:**

This paper proposes a new and important issue about diffusion models, namely direct finetuning a timestep-distilled diffusion model without apparent performance degradation. Since finetuning a timestep-distilled diffusion model with a naive diffusion loss will result in degraded generated images, this paper proposes a so-called pairwise sample optimization algorithm that increases a relative likelihood margin between the training images and reference images. The authors conduct several experiments on different acceleration methods, including DMD2, Turbo, and LCM on different finetune tasks, i.e., humen preference tuning, style transfer, and customized image generation. Experimental results can demonstrate the effectiveness of the PSO

**Strengths:**

1. This paper addresses a very important and practical issue in the diffusion-model field.
2. The proposed pairwise sample optimization (PSO) algorithm is novel and has thereotical derivation to increase its rigorism。
3. This paper considers both offline and online setting, whichi can unifies various prior works.
4. Experimental studies are conducted on different acceleration methods and applied on several downstream finetune tasks, which make this PSO more persuasive.

**Weaknesses:**

1. The analysis and description of the proposed PSO are slightly incomplete. For instance:
- This paper doesn't explain why "directly adopting the DDPM  formulation and minimizing the diffusion objective $||\epsilon\theta(x_{t_n}^\tau) − ε||^2$ leads to blurry."
- Why you choose to "recast the optimization as maximizing the relative likelihood between the target and reference samples?'' I think this is the most crucial insight of this paper.
- How to chooose the "pre-sampled reference image samples''?
2. The experimental results in Table 1 are a little confusing. Why do you choose "SDXL-DPO + DMD2-LoRA'' as a compared baseline? Use DMD2-LoRA based on a SDXL-DPO model? Why not compare with "SDXL-DMD2-LoRA  + DPO'' by using the distilled SDXL-DMD2 as base model?
3. Some typos: Figure 1(b) -> Figure 1(c) in Line 107; the extend it to -> then extend it to in Line 148; no explanation for $p_{pre}$ in Eq (1) (maybe $p_{\theta_{pre}}$)

**Questions:**

Please refer to Weakness

---

> ### Author Response · Authors · 2024-11-25
> **Response to Reviewer B3DX**
>
> Thanks to the reviewer for acknowledging our contribution. We provide our response to the questions below.
>
> **Q1: More Analysis on PSO.**
>
> * **Why directly minimizing the diffusion objective leads to blurry.**
> The diffusion objective, $||\epsilon_\theta'(x_t, t)-\epsilon||^2$, trains the diffusion model $\epsilon_\theta'$ to predict the score as, $\nabla_x\log p(x_t)=- \epsilon_\theta'(x_t, t) / \sigma_t$, which be used to solve probability flow ODE(PF ODE) as the diffusion model sampling, $\frac{dx_t}{dt} = f(t)x_t + \frac{g^2(t)}{2\sigma_t}\epsilon_\theta'(x_t, t)$. Solving the reverse ODE, i.e., sampling from multi-step diffusion models, usually requires 50 ODE solver steps. That is, the model trained with the diffusion objective is used for 'local' prediction(small timestep range, e.g., $t=975$ to $t=950$) on PF ODE. On the other hand, the trained timestep-distlled model $\epsilon_\theta$ is trained to directly predict the endpoint $x_0$ of the PF ODE from arbitrary point $x_t$, which is a 'global' prediction, and obtain the few-step generation ability as a result. In this way, directly tuning the distilled model using the original diffusion objective will distort its 'global' prediction ability from $x_t→ x_0$, and resulting in blurry results, which is shown in Figure 1.
>
> * **Why recast the optimization as minimizing the relative likelihood.** We choose to recast the optimization as maximizing relative likelihood between target and reference samples to avoid directly tuning the distilled model with 'local' score matching objective, as stated above, which leads to blurry few-step generation results. Inspired by some initial human preference tuning experiments with pairs of data, we find that by instead maximizing the relative likelihood ratio $\log\frac{p_θ(x^τ_0|c)}{p_θ(x^ρ_0|c)}$, we can preserve the few-step generation ability while effectively steering the generative distribution toward desired targets. Thus, we design PSO that maximizes the relative likelihood of pairwise data, which is further validated in other tuning experiments besides human preference tuning.
>
>
> * **How to choose pre-sampled images.** We select the pre-sampled reference images from the model distribution, which contributes in PSO to preserve the few-step generation ability. Specifically, in human preference tuning, we directly adopt the negative samples in the pickapicv2 dataset as the reference images, as most of them are sampled with SDXL, which is close to the generative distribution of the distilled models. While in style tuning, we pre-sample the reference images from the distilled model using the prompts in the style transfer dataset.
>
>
> **Q2: Question on Baselines selected, why SDXL-DPO + DMD2 LoRA, rather SDXL-DMD2 + DPO?**
>
> Thanks to the reviewer for bringing this up. We would like to compare to the baseline where the multi-step diffusion model is tuned for human preference first (SDXL-DPO), and then distilled for few-step generation (DMD2). This baseline would be too costly to obtain (i.e., 3840 A100 GPU hours in Line 52), and we choose the DMD2 LoRA applied to SDXL-DPO (SDXL-DPO + DMD2 LoRA) as an efficient proxy. Meanwhile, as stated in Line 244, our Offline-PSO has a very similar formulation to Offline Diffusion-DPO. Thus, the required baseline SDXL-DMD2-LoRA + DPO can be seen as Offline-PSO results in the Tables (as discussed in Line 95), which shows effectiveness in tuning timestep-distilled diffusion models, and yet is not as effective as the proposed Online-PSO in human preference tuning.
>
>
> **Q3: Typos.**
>
> Thanks to the reviewer for pointing them out. We have revised that in the revision.

---

### Meta-Review · Area_Chair_uvUB · 2024-12-09

**Metareview:**

All reviewers agree to accept the paper. Reviewers appreciate the novel task, the innovative method, and extensive experiments. Please be sure to address the reviewers' comments in the final version.

**Additional Comments On Reviewer Discussion:**

All reviewers agree to accept the paper.

---

### Decision · Program_Chairs · 2025-01-22

Accept (Poster)